Dear DMLR reviewers and editors,

This PDF groups the final version of the following chapters:

- Chapter 10: Competition platforms
- Chapter 11: Hand-on tutorial on how to create your own challenge or benchmark
- Chapter 12: Special designs and competition protocols
- Chapter 13: Practical issues: Incentives, community engagement and costs

Thank you in advance. Have a nice read!

Adrien Pavão

# Competition platforms

**Andrey Ustyuzhanin**                    ANDREY.USTYUZHANIN@CONSTRUCTOR.ORG
*Constructor University, Bremen,*
*Campus Ring 1, 28759, Germany*

**Harald Carlens**                                     HARALD@MLCONTESTS.COM
*ML Contests*

**Reviewed on OpenReview:** *https://openreview.net/forum?id=aATSw1zwOV*

## Abstract

The ecosystem of artificial intelligence contests is diverse and multifaceted, encompassing several platforms that each host numerous competitions and challenges annually, alongside many specialized websites dedicated to individual contests. These platforms manage the overarching administrative responsibilities inherent in orchestrating contests, thus allowing organizers to allocate greater attention to other aspects of their contests. Notably, these platforms exhibit considerable variety in their features, economic models, and communities. This chapter conducts an extensive review of the leading services in this space and explores alternative methods facilitating the independent hosting of such contests. We provide hints and tips on choosing the right platform for your challenge at the end.

**Keywords:** competition platform, challenge hosting services, service comparison

## 1 Platforms for AI contests

The majority[1] of AI contest organisers use a third-party platform to host their contest rather than building and maintaining their own infrastructure.

The choice of platform is driven by various considerations. Before we introduce these, and the role we expect a platform to fulfil, it's helpful to return to a definition of the types of contests we're considering. We can ground our expectations of a platform in the Common Task Framework (Donoho, 2017) (CTF), which lays out the key ingredients of an AI challenge:

1. A publicly available training dataset, involving a list of feature measurements and a class label for each observation;

2. A set of enrolled competitors whose common task is to infer a class prediction rule from the training data;

3. A scoring referee, to which competitors can submit their prediction rule. The referee runs the prediction rule against a testing dataset which is not made available to competitors. The referee objectively and automatically reports the score achieved by the submitted rule.

---

1. 317 of 367 contests in 2023 were hosted on a third-party platform (Carlens, 2024). The universe of contests considered here includes only those with meaningful prize money (over $1,000) or a conference affiliation.

While these "ingredients" are somewhat specific to supervised learning problems, it is not too difficult to see how they would generalise to other fields - such as reinforcement learning style challenges, where data sets are replaced with environments. They also generalise to our broader definition of "contests", which includes these measurable challenges as well as competitions where performance is evaluated by a panel of judges[2]. In the case of a subjectively-judged competition, the judged output generally includes a written document or working prototype in addition to, or instead of, a simple prediction rule. From the above ingredients we can get a list of responsibilities, to be shared between organisers and platforms:

- **Design**: framing a problem in a way that is amenable to a CTF-style contest

- **Data**: gathering and cleaning data for training and test datasets

- **Discovery**: notifying potential participants

- **Admin**: publishing rules and making training data available

- **Engagement**: enabling participants to be productive and to collaborate

- **Scoring**: accepting submissions, evaluating them, and updating leaderboards

- **Dissemination**: sharing insights from submissions and contest outcomes

It is possible for all of these responsibilities to be undertaken by the contest organisers, or for them all to be outsourced to a platform, but in most cases the responsibilities are shared. The decision of which responsibilities are to be outsourced is of primary importance in the choice of platform, as some are better suited to certain responsibilities than others[3]. Secondly, the exact requirements for each responsibility will further determine the choice of platform [4]. The remaining components of platform choice come down to budget, familiarity with different platforms, and geographical considerations.

---

2. Throughout this article, we aim to conform to the nomenclature used by the other chapters in the "AI Competitions and Benchmarks" book, of which this article will make up Chapter 10. The common definitions for contest, competition, challenge, and benchmark are as follows:

   Contest: A contest is an event created by organizers, governed by rules, and directed to a group of participants, offering the opportunity to win an award or a prize.

   Competition: A competition is a skill-based scientific contest, with a limited time duration, involving the submission of proposals, project propositions, project outcomes, and/or prototypes that are evaluated by a panel of judges.

   Challenge: A challenge is a skill-based scientific contest, with a limited time duration, ending by a total ranking of participants according to a pre-defined scoring metric, and the selection of winners.

   Benchmark: A benchmark refers to on-going evaluations of methods or models in well-defined conditions, for the purpose of making standardized comparisons.

   There will be occasional inevitable exceptions, including where products use names that conflict with our definitions - e.g. "Kaggle Competitions".

3. For example, Codabench is able to fulfil most of these, but does not provide support with design or data preparation.

4. For example, is the contest aiming to reach a broad audience, or is it targeted at a niche research community who can all be reached through a single mailing list? Is it a straightforward supervised learning problem, or does it incorporate adversarial elements or reinforcement-learning style environments for scoring?

With these responsibilities in mind, in the next section we lay out more detailed criteria and review the main features of several leading platforms:

- AIcrowd (Mohanty et al., 2017),

- Codabench (Xu et al., 2022) by Université Paris-Saclay,

- CodaLab (Pavao et al., 2023) by Université Paris-Saclay,

- DrivenData (DrivenData, 2014),

- EvalAI (Yadav et al., 2019) by CloudCV,

- Kaggle (Goldbloom and Hamner, 2010) by Alphabet Inc,

- Tianchi (Group, 2014) by Alibaba,

- Zindi (Zindi, 2018).

This list is not intended to be comprehensive, and is focused on generalist platforms with active communities as of the end of 2022. We give a separate overview of several non-generalist platforms that target specific domains or follow a complementary pattern that doesn't strictly adhere to the CTF setup, as well as some non-English language platforms.

## 2 Platform comparison criteria

We outline the main characteristics that we use for comparison, which are provided roughly in order of the responsibilities listed above.

**Design support:** platforms vary in the amount of support they are able to provide to organisers in designing a contest. Here we are defining the "design" process to cover the initial problem formulation, decisions around the train/test split, and choice of evaluation metrics. This tends to be most important for companies with little in-house data science expertise, and not so relevant for researchers with specific problems in mind.

**Data support:** some platforms help organisers gather and clean data, as well as transforming it into a format that is convenient for participants to use.

**Registered users:** the total number of users registered on a platform gives a good indication of the size of the audience that can be reached. This is particularly important for organisers looking to reach a broad audience, or to reach participants who are not already familiar with the problem area or organiser.

**Code sharing:** some platforms allow structured code-sharing through notebooks which can be hosted and executed on the platform. Others allow participants to embed their solution as an external notebooks or code repositories. This functionality allows other participants to easily reproduce and build on others' solutions. Open community collaboration in this way can be a valuable feature for complicated or novel challenges.

**Submission code evaluation:** the most straightforward way to run a challenge is to ask participants to submit a set of predictions, and compare those against some "ground truth" values using a loss metric. Many platforms allow for challenges where participants submit code that is then run on the platform side to generate predictions against unseen

data. This allows organisers to do things like impose compute budget constraints on submissions, and vet submissions for compliance with the rules. It also changes the nature of the challenge, since participants have less knowledge about the distribution of test set examples than they do in the case where they have access to test set features. Most platforms that support code submissions can support it in any language, though support for Python and R tends to be better than for other languages.

**Custom metrics:** some platforms or offerings are able to support only "common" metrics like mean squared error or cross-entropy loss. The ability to implement custom metrics is important for many challenges, especially those looking to capture particular trade-offs. Some platforms allow choosing just one among many predefined metrics; some allow for custom implementations. Some platforms charge an additional cost for implementing non-standard metrics.

**Staged contests:** contests from within a niche domain can initially look inscrutable to the wider community, and it can help to split the contest into smaller chunks of gradually increasing complexity. A preliminary trial stage can also help to mitigate risks of data leakage. In addition to this, it has been shown that pre-filtering participants in a trial run can help reduce over-fitting on winner selection. (Pavao et al., 2022)

**Private evaluation:** Data privacy is a sensitive issue. Some platforms allow participants' solutions evaluation using an organizer's dedicated machines. With the help of such a feature, one can set up a challenge without needing to share restricted code or datasets with anyone, even with platform owners. This feature can also be used to support unusual evaluation procedures - for example, those needing to run on specific hardware managed by organisers, or on a physical robot in a lab.

**Reinforcement Learning (RL) evaluation:** running participants' RL agents on the platform's side is inherently more complex than running a metric evaluation script across a vector of predictions, and not all platforms support this. Computational cost for these types of challenges is not only often higher than for supervised learning problems, but also more unpredictable - since RL evaluation episode lengths can depend on the success of an agent. Supporting multi-agent environments or tournament-style evaluations are an additional challenge, and we do not evaluate this ability in our analysis.

**Judging panel:** some contests do not use a simple scoring metric, and instead are evaluated by a panel of judges. These competitions sometimes incorporate more open-ended elements of data exploration and discovery, or require participants to develop prototype solutions or products which are not easily evaluated in an automated and measurable way.

**Human-in-the-loop (HITL) evaluation :** some contests do not have ground-truth labels in the data, and require large-scale human evaluation for comparison. For example, a dialogue bot evaluation requires communication with a living person. Some platforms enable the use of human-evaluation platforms, such as Amazon Mechanical Turk (see below).

**Run for free:** most platforms charge a fee. The exact cost usually depends on the range of services offered. Some platforms offer a free "self-service" offering, allowing organisers to set up a completely self-managed contest.

**Open-source:** for some platforms, the code that runs them is open-source. In most cases a platform fulfils a service, and organisers do not have an interest in changing the platform's functionality. However, being able to access a platform's source code can help organisers assess the pace of development on the platform, verify details of the platform

| Criteria | AIcrowd | Codabench | CodaLab | DrivenData | EvalAI | Kaggle | Tianchi | Zindi |
|---|---|---|---|---|---|---|---|---|
| Design support | ✓ | - | - | ✓ | - | ✓ | ✓ | ✓ |
| Data support | ✓ | - | - | ✓ | - | ✓ | ✓ | ✓ |
| Registered users | 140k+ | 5k+ | 55k+ | 100k+ | 40k+ | 16m+ | 1.4m+ | 70k+ |
| Code sharing | ✓ | ✓ | ✓ | ✓[6] | ✓ | ✓ | ✓ | ✓ |
| Code evaluation | ✓ | ✓ | ✓ | ✓ | ✓ | ✓ | ✓ | - |
| Custom metrics | ✓ | ✓ | ✓ | ✓ | ✓ | ✓ | ✓ | ✓ |
| Staged contests | ✓ | ✓ | ✓ | ✓ | ✓ | - | ✓ | - |
| Private evaluation | ✓ | ✓ | ✓ | - | ✓ | - | ✓ | - |
| RL-friendly | ✓ | ✓ | ✓ | - | ✓ | ✓ | - | ✓[7] |
| Judging panel | ✓ | - | - | ✓ | - | ✓ | ✓ | ✓ |
| HITL evaluation | ✓ | - | - | - | ✓ | - | - | - |
| Run for free | - | ✓ | ✓ | - | ✓ | ✓[8] | ✓ | ✓[9] |
| Open-source | - | ✓ | ✓[10] | - | ✓[11] | - | - | - |
| Established | 2017 | 2023 | 2013 | 2014 | 2017 | 2010 | 2014 | 2018 |

Table 1: Platform overview

evaluation mechanics, or allow organisers to run their instance on their own premises for a local event with private datasets. It also allows organisers to add features to the platform themselves.

## 3 Platform Comparison

An overview of platforms as measured by the criteria above is presented in Table 1[5]. Features which we were able to verify as being supported are marked as ✓, and where possible these were confirmed with the team running the platform. In some cases where we could not find public documentation of a feature and we did not receive any response from the platform operators, it is possible that we have incorrectly marked features as unavailable. Estimates for the number of users and typical number of entries reflect activity in 2023 (Carlens, 2024).

Here are some highlights of the platforms included in the comparison.

**AIcrowd** started as a research project at EPFL, and has since run a large variety of competitions. It has hosted several official NeurIPS competitions including many reinforcement learning challenges.

**Codabench** is an open-source platform, with an instance maintained by Université Paris-Saclay. Anyone can sign up and host or take part in a contest. Free CPU resources are available for inference, and organisers can supplement this with their own hardware. Codabench is friendly to a variety of challenges: from online data science classes/hackathons to contests affiliated with leading conferences, and can also be used for ongoing benchmarks. Codabench is suitable to organisers who have a clear idea of the contest they want to run, and can be self-sufficient when it comes to technical and marketing aspects.

---

5. One of the authors maintains an updated version of this table at `https://mlcontests.com/platforms/`.

**CodaLab** is the predecessor of Codabench, and is maintained by the same team. Where possible, the team recommends that organisers use the newer Codabench platform. The only exception to this is for challenges which require ranking participants based on an aggregate of multiple different scores, a feature which is supported by CodaLab but not yet by Codabench as of the time of writing.

**DrivenData** focuses on running contests with social impact, and has run competitions for NASA and other organisations. DrivenData stands out for its thorough reports detailing participants' approaches, and permissively licensed solution code publication[12].

**EvalAI** is built by a team of open source enthusiasts working at CloudCV, a consulting company that aims to make AI research reproducible and easily accessible. With the platform's help, they reduce the entry barrier for research and make it easier for researchers, students, and developers to design and use state-of-the-art algorithms as a service. It is known for running many competitions involving human-in-the-loop evaluations.

**Kaggle** was acquired by Google in 2017 and has the largest community of all the platforms, with over 16 million registered users. As well as hosting contests, Kaggle allows users to host datasets, notebooks, and models. Kaggle's progression system [13] provides additional incentives for users users to compete, collaborate, share code, and contribute to community discussions. It is possible to run a "Community Competition" for free, with limitations around discoverability, evaluation metrics, and participant incentives.

**Tianchi** is a platform run by Alibaba, including running kernels and earning points. Contests can quickly gain several thousand participants. The primary audience is Chinese, though many contests also include English documentation.

**Zindi** is focused on connecting organisations with data scientists in Africa. As well as online contests, Zindi also runs in-person hackathons and community events.

The table above only lists a few of the largest existing platforms. These are some other general-purpose platforms worth exploring:

**bitgrit**[14] is an AI contest and recruiting platform founded in 2017, with over 55,000 registered users.

**Hugging Face** launched its Competitions[15] platform in February 2023, alongside its well-established Model Hub and widely-used open source machine learning repositories.

**Humyn.ai**[16] hosts contests as well as facilitating deeper engagements between businesses and its user-base of data scientists.

There is a long tail of platforms, and we expect that there are other relevant platforms which are as yet unknown to us.

## 4 Non-English language platforms

The comparison above is focused on English-language platforms. While the authors are less familiar with platforms in other languages, this section is an attempt at covering platforms in regions where the main common language of their audience is not English. As already

---

12. https://github.com/drivendataorg/competition-winners
13. https://www.kaggle.com/progression
14. https://bitgrit.net/competition/
15. https://huggingface.co/competitions
16. https://humyn.ai

mentioned, the most notable **Chinese** platform is Tianchi. Other Chinese platforms worth mentioning are: Data Castle[17], Kesci[18], Bien Data[19], and Data Fountain[20]. The **Japanese** platform Signate[21] and the company behind it collaborate with industries, government agencies, and research institutes in various domains to resolve social issues. The **Russian** community, Open Data Science[22] runs contests, as well as including organizing events and finding joint projects for researchers, engineers, and developers around Data Science. Other Russian websites listing contests include DS Works[23] and Yandex Cup[24]. All these platforms above have a reasonably developed community; however, to join those, one needs to be fluent in the corresponding language.

## 5 Domain-specific platforms

Several platforms host regular challenges on domains in a specific branch of science or industry, or within a more narrow scope than the Common Task Framework. We list a few examples here.

**DREAM Challenges**[25] has been running biomedical challenges since 2006, with now over 30,000 registered users.

**Grand Challenge**[26] is a platform for the end-to-end development of machine learning solutions in biomedical imaging. It has successfully run over a hundred challenges, and allows researchers to host custom algorithms that can be used for performance assessment on new datasets and crowd-sourcing activities called *reader studies.*

**Makridakis Open Forecasting Center (MOFC)**[27] conducts research on forecasting, and has been running the "M Competitions", a series of forecasting challenges, since 1987. The most recent M Competition was the M6 Financial Forecasting Competition, running from 2022 until 2023.

**NASA Tournament Lab**[28] (NTL) facilitates the use of crowd-sourcing to tackle NASA challenges. NASA's researchers, scientists, and engineers have launched numerous crowd-sourcing projects through the NTL, seeking novel ideas or solutions to accelerate research and development efforts in support of the NASA mission. The NTL offers a variety of open innovation platforms that engage the crowd-sourcing community to improve solutions for specific, real-world problems being faced by NASA and other Federal Agencies.

**Numerai**[29] is a fund that draws its strategy from crowd-sourced predictions submitted to regular tournaments. Participants aim to predict stock market movements from obfuscated data. Numerai states that it has paid out over $48m to its data scientist collab-

---

17. `https://challenge.datacastle.cn/v3/cmptlist.html`
18. `https://www.kesci.com/`
19. `https://www.biendata.xyz/`
20. `https://www.datafountain.cn/`
21. `https://signate.jp/`
22. `https://ods.ai/`
23. `https://dsworks.ru/`
24. `https://yandex.com/cup/ml/`
25. `https://dreamchallenges.org`
26. `https://grand-challenge.org/`
27. `https://trustii.io`
28. `https://www.nasa.gov/coeci/ntl`
29. `https://numer.ai/`

orators. It is worth noting that reward eligibility in Numerai tournaments requires staking Numerai's NMR cryptocurrency token, exposing participants to potential losses, unlike most other platforms listed here.

**Onward**[30] is a platform run by Shell, which is focused on enabling innovation in the energy sector. Many of the contests run on this platform so far have been targeted at solving specific business problems, and so the winning solutions are not generally shared publicly.

**Solafune**[31] was founded in 2020, and focuses on competitions using satellite and geospatial data.

**Trustii**[32] is a platform established in 2020, primarily focused on healthcare competitions. Winners' code and solutions are shared on GitHub.

**Unearthed**[33] is a platform that hosts contests aimed at making the energy and resources industry more efficient and sustainable. Challenges often involve a mixture of domain knowledge and data science skills.

## 6 Alternative approaches and adjacent services

The platforms above are the most notable ones implementing contests broadly in line with the Common Task Framework (Donoho, 2017). However, they are far from the only options for collaborative research. Below is a list of platforms and services that rely on different assumptions and implement interaction protocols that turn out to be suitable for research goals in some scientific domains, or that can aid in running CTF-style competitions in a role other than a competition platform.

**Amazon Mechanical Turk (AMT)**[34]: a marketplace for completion of virtual tasks that require human intelligence. Businesses or academic researchers regularly use it to label data that can later

**DataCamp**[35] a data science education platform which hosts occasional competitions targeted at beginners.

**Dynabench**[36]: a platform for dynamic data collection and benchmarking that aims to address issues with static benchmarks through human-in-the-loop benchmarking.

**InnoCentive**[37]: is an innovative hub for a new kind of problem-solving. It describes the framework of "Challenge Driven Innovation" (CDI) that helps reformulate a task or opportunity at hand into a series of modules or challenges addressed later by a network of participants. CDI framework is much broader than CTF. Thus Innocentive enjoys various challenges, including Brainstorming, Design, Prototyping, and Algorithm development. The platform has been around for over a decade. It links over half a million solvers and spans dozens of industries.

---

30. https://thinkonward.com
31. https://solafune.com
32. https://trustii.io
33. https://www.unearthed.solutions
34. https://www.mturk.com/
35. https://www.datacamp.com
36. https://dynabench.org/
37. https://www.innocentive.com/

**LMSYS Chatbot Arena**[38] is a crowd-sourced open platform for evaluating large language models through pairwise comparison.

**Google Colab**[39]**:** a hosted notebook solution with support for CPU/GPU/TPU accelerators and sharing via GitHub or Google Drive, Google's Colab service enables interactive code-sharing and eases reproducibility. It significantly lowers the bar for researchers to interact with code or libraries that are not within their domain of expertise, by enabling them to run and edit code without needing to worry about maintaining environments or installing libraries. It can be a useful place for organisers to share code examples with potential participants, allowing them a frictionless way to explore a contest.

**ML Experiment Tracking Tools:** Tools like MLflow[40] (open source), W&B[41], Comet[42], Neptune[43] enable distributed research teams to easily share their experimental results within their team or to a public audience. These can serve as a more useful record of experimental results than local tools like TensorBoard or simple text-based logs. be used for training ML algorithms. AMT has been around for more than 15 years. Major companies like Google and Microsoft have similar versions of such marketplaces.

**ML Collective**[44]**:** (MLC) is an independent, nonprofit organization with a mission to make research opportunities accessible and free by supporting open collaboration in machine learning (ML) research. Jason Yosinski and Rosanne Liu founded MLC at Uber AI Labs in 2017 and, in 2020, it moved outside Uber. The group aims to build a culture of open, cross-institutional research collaboration among researchers of diverse and non-traditional backgrounds. Thus, the outcome of the cooperation is the natural growth of participating researchers through discussion and publishing process participation. As of mid-2022, the community is more than 3 thousand ML researchers sharing collaborative research values.

**ML Contests**[45]**:** is primarily a contest discovery platform, with a listing page that shows currently active contests across many platforms. Organisers can add their contest to the listing page for free. Alongside this, ML Contests also publishes research on competitive machine learning.

**MLCommons**[46]**:** an AI engineering consortium, bringing together groups from industry and academia to foster collaboration and set standards. As well as maintaining the MLPerf benchmarks, MLCommons runs working groups on benchmarks, AI safety, data, and research.

**OpenChallenges**[47]**:** a centralised hub for biomedical challenges across various platforms, maintained by Sage Bionetworks.

**OpenML**[48]**:** an online machine learning platform for sharing and organizing data, machine learning algorithms, and experiments. Thus they have created a service that allows

---

38. `https://chat.lmsys.org/`
39. `https://colab.research.google.com/`
40. `https://mlflow.org/`
41. `https://wandb.ai/`
42. `https://www.comet.com/`
43. `https://neptune.ai/`
44. `https://mlcollective.org/`
45. `https://mlcontests.com`. Note: ML Contests is maintained by one of the authors of this article.
46. `https://mlcommons.org/`
47. `https://openchallenges.io`
48. `https://www.openml.org/`

running an algorithm across several datasets and systematically comparing its performance. While there are no private leaderboards, every check is systematically performed via system API and protocol. Thus new experiments are immediately compared to state of the art without always having to rerun other people's experiments. The recent development of OpenML involves the design of an AutoML evaluation framework for a broad spectrum of datasets.

**Papers With Code**[49]: organizes access to scientific papers from the leading Machine Learning conferences and links to known implementations of the methods described in such articles. The service also compares different methods of solving several tasks in the form of a leaderboard where entries are linked to particular implementations. The diversity of such leaderboards has grown immensely in the past few years. With the help of this platform, one can find the most current state of the art to the problem of interest and read details of the method in the companion paper.

**Seasonal events:** there are many yearly data analysis events organized around the world. Usually, those are hosted by universities and attract quite a significant number of participants. International Data Analysis Olympiad (IDAO)[50] is just a single example among many others[51],[52]. IDAO has engaged several thousand participants across almost a hundred countries each year since 2019. Besides reaching out to a big community, organizers usually run a series of events, including online and offline interactions with the participants.

**Zooniverse**[53]: Zooniverse builds a community of people interested in contributing their efforts and intelligence to scientific research advances. It provides participants with unlabelled datasets from various scientific branches: biology, climate, history, physics, etc. Those datasets require human intelligence to label and understand the scientific assumptions of the domain and phenomena presented. Participation in real-science research can motivate people quite significantly. In some cases, discussions between scientists and Zooniverse participants lead to new scientific discoveries (Clery, 2011).

**Other:** There are many different venues for interactions between science and citizens. In his book "Reinventing Discovery: The New Era of Networked Science" (Nielsen, 2020), Michael Nielsen gives a good overview. An interesting example of such interaction is the design of a network of micro-prediction agents that follow a specific question-answering protocol. Authors of those agents get rewards for providing correct answers. Such protocol incentivizes the participants to come up with better algorithms and suitable external data sources (Cotton, 2019). A broader list of citizen-science projects is, of course, available at Wikipedia (wik).

## 7 Independently hosted contests

As we've seen, most organisers choose to host their contests on a platform. However, others have shown that it's still possible to "self-host" contests. Here we give a few brief examples of independently hosted contests.

---

49. https://paperswithcode.com/
50. https://idao.world
51. Data Mining Cup, https://www.data-mining-cup.com/
52. ASEAN Data Science Explorers https://www.aseandse.org/
53. https://www.zooniverse.org/

**MIT Battlecode**[54] is an annual competitive real-time strategy game where players need to write code to manage a robot army. The first iteration took place in 2003, predating all currently active contest platforms. Anyone can participate, but only student teams (from any university) are eligible for prizes. Recent sponsors include game studios and quantitative trading firms. MIT students participating in Battlecode are eligible for credits, as it is a registered course.

**Real Robot Challenge** (Bauer et al., 2022)[55] is a contest involving dexterous manipulation tasks using robot hands. Evaluation takes place on physical robots. Participants are provided with software simulation environments to train their policies, and are able to submit their policies for physical evaluation. The organising team had to do significant development work in order to be able to accept submissions to run on their physical robots, and they decided to self-host the whole contest since the additional work to build their own leaderboard was deemed easier than integrating with an existing platform.

**The ARCathon**[56] is an ongoing abstract reasoning benchmark that spun out of the 2020 Kaggle Abstraction and Reasoning Challenge. It maintains the same closely-guarded test set, and their website provides tools for exploring the training set manually as well as crowdsourcing new training examples.

**The Humanoid Robot Wrestling Competitions**[57] are a series of simulated robotics challenges. The organisers built their own leaderboard management framework on top of GitHub Actions, enabling anyone with a GitHub account to take part in the challenge. Evaluations are run automatically on a dedicated server managed by the organisers whenever a competitor pushes a code change to their GitHub repository. Participants' code can stay hidden from other participants; participants just need to add the challenge organiser's GitHub account as a collaborator on their repository. The organisers helpfully shared their challenge template[58] under a generous open-source licence, enabling others to run challenges like this with minimal additional setup.

**The Vesuvius Challenge**[59] aims to decipher millenia-old carbonised papyrus scrolls by using computer vision algorithms on high-resolution non-invasive x-ray scans. Alongside the main prizes for reading characters or passages from the scrolls, the organisers offer various prizes for preliminary progress, and contribtions to the community through tool-building or information sharing. The organisers hosted an image segmentation challenge on Kaggle for the subproblem of ink detection, but most of the prizes require submission through a Google form.

## 8 Choosing the right platform

Given the set of platforms available, choosing the one best suited to a particular competition or challenge is not trivial. We hope that table 1 can be a helpful resource for contest organisers. In addition to this, we can provide some general advice.

---

54. `https://battlecode.org`
55. `https://real-robot-challenge.com`
56. `https://lab42.global/arcathon/`
57. `https://webots.cloud/competition`
58. `https://github.com/cyberbotics/competition-template`
59. `https://scrollprize.org/`

For companies with limited in-house data science expertise or tech resources, it makes sense to choose a platform which offers support with challenge design and data preparation. While these platforms can require a larger budget than alternative options, they are often able to leverage their existing significant user-base to engage desirable and capable competitors, resulting in more and higher-quality submissions than might otherwise be possible. This reduces the pressure on organisers to promote the contest themselves.

Contest organisers with a limited budget will generally have to take on the challenge design and data preparation work themselves. In these cases, unless an additional contest sponsor can be found, using a platform with free contest hosting options will likely be desirable. In order to aid with discoverability on the free hosting options - helping potential participants find the contest - organisers might want to try to get their competition mentioned in relevant newsletters, or submit their contest to a contest listing site like ML Contests if they are trying to reach a broad audience.

Teams organising contests with particular requirements - reinforcement learning environments, data privacy restrictions, or human-in-the-loop evaluation - are more restricted in their choice of platforms than "vanilla" supervised learning contests. It's worth noting that even if platforms don't officially list certain features, sometimes they are able to accommodate additional requirements - so it can be worth having an exploratory conversation before ruling out any platforms, as long as sufficient budget is available to compensate platforms for any additional development they might need to do.

Contests targeted at niche communities might benefit from the relevant exposure they would get on a domain-specific platform (see section 5). Similarly, contests targeting participants with certain language skills or located in particular geographic areas might take this into account when choosing a platform (see section 4).

Only organisers of the most idiosyncratic contests or those with significant in-house resources would likely find it preferable to run a contest without making use of any platform. We mention some examples of these in section 7.

## 9 Conclusion

This chapter presents an overview of the most popular AI contest platforms. It gives a summary of each of the platforms, introduces key criteria for platform comparison, and uses these to provide a simple comparison table that we hope will be a useful reference for any contest organiser looking to find the most suitable service for running their contest and maximising its potential impact.

## Acknowledgments and Disclosure of Funding

The work presented in this book chapter was undertaken as a community collaboration and did not receive any external funding.

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

# Hands-on tutorial on how to create your own challenge or benchmark

**Adrien Pavão**                                                    ADRIEN.PAVAO@GMAIL.COM

*LISN, CNRS*
*Université Paris-Saclay*
*France*

**Reviewed on OpenReview:** *https://openreview.net/forum?id=aATSw1zwOV*

## Abstract

Organizing a challenge allows you to crowd-source the most difficult machine learning problems. It is also an excellent way to learn data science. By following this short hands-on tutorial, you can create your first competition or benchmark — as early as today! In this chapter, we give you everything you need to implement, concretely, your own online competition or benchmark. We do not address other practical issues such as finding sponsors or communicating about the event; this is discussed in chapter 13.

**Keywords:**  tutorial, CodaLab, Codabench

## 1 Introduction

In this chapter, you will learn how to organize, a challenge or a benchmark on the two platforms *CodaLab Competitions* and *Codabench* (Figure 1). This tutorial is divided into three parts: we first review the aspects shared by both platforms (Section 2), then the *CodaLab Competitions* platform specificity (Section 3), and finally the *Codabench* platform specificity (Section 4).

**CodaLab Competitions** Pavao et al. (2022) is an open-source web platform hosting data science and machine learning competitions. This means that you can set up your own instance of it, or use the main instance on codalab.lisn.fr. *CodaLab* puts an emphasis on science and each year hundreds of challenges are organized on it, pushing the limits in many areas: physics, medicine, computer vision, natural language processing or even machine learning itself. Its flexibility allows hosting challenges on a wide variety of tasks! The only limit is your imagination.

**Codabench** Xu et al. (2022) is another project, free and open-source as well, following the steps of *CodaLab Competitions*. It can be seen as an upgrade of it, using more recent technologies, and with an emphasis on benchmarks. This emphasis on benchmark is enforced by some features, such as the possibility of filling a leaderboard with a single user account. The public server is codabench.org.

These two platforms are well suited for a tutorial, given their flexibility and the fact that they are open-source and free to use. **Once you have an account, you can already publish your first competition or benchmark!**

Figure 1: Sources: codalab.lisn.fr, codabench.org

## 2 General aspects

### Inside the competition bundle

To create a machine learning challenge or benchmark on these platforms, all you need to do is to upload a **competition bundle**. A competition bundle is a ZIP file containing all the pieces of your competition: the data, the documentation, the scoring program and the configuration settings. To customize your competition, you can simply change the files contained in the template bundle before uploading it. Note that every aspects of the competition (settings, data, etc.) can still be edited after the upload. Let's have a closer look at what's inside the bundle.

**The competition.yaml file** defines the **settings** of your challenge. The title, description, logo, dates, prizes, Docker image[1], leaderboard structure and so on. All possible settings are documented in the Wiki.

**The documentation files**, either *HTML* or *Markdown* files, define the various pages that participants can see when going to your competition. Use them to provide the documentation and the rules, as well as any information you find important. You can of course select your own **logo** for the competition by replacing the "logo.png" file.

**Data**. If you are designing a machine learning problem, it is likely that you have data. The **public data** folder is for the data fully accessible by participants, **input data** is accessible by participants' submitted code, and the **reference data** folder is for storing the ground truth information (e.g. the labels from the testing set) which is kept hidden to the participants, making it only accessible by the scoring program, as explained later in this section. You can either use a data format provided in a competition example or a new one that fits well with your competition. To ensure compatibility, you will need to update the scoring program — we will talk about it in the next section. *If your problem does not involve data, don't worry! CodaLab is flexible and allows you to define any kind of problem (e.g. reinforcement learning tasks).*

**The ingestion and scoring programs** are the critical pieces of your competition bundle since they define the way submissions will be executed and evaluated respectively. If you want to allow only **result submissions**, then you only need the scoring program; the ingestion program is useful for **code submissions**. Figure 2 shows the interactions between the submissions (results submissions or code submissions), the programs and the leaderboard.

- The **ingestion program** defines how to train the models and save their predictions.

- The **scoring program** defines how to compare the predictions with the ground truth and computes a score.

---

[1]The Docker image is the environment in which all submissions will be run, allowing to precisely control the evaluation procedure. It can be referred by its DockerHub name.

These programs evaluating submissions can be customized by the organizers to adjust them to adapt to any competition protocol. While they are written in Python in the templates provided, they can be written in any programming language.

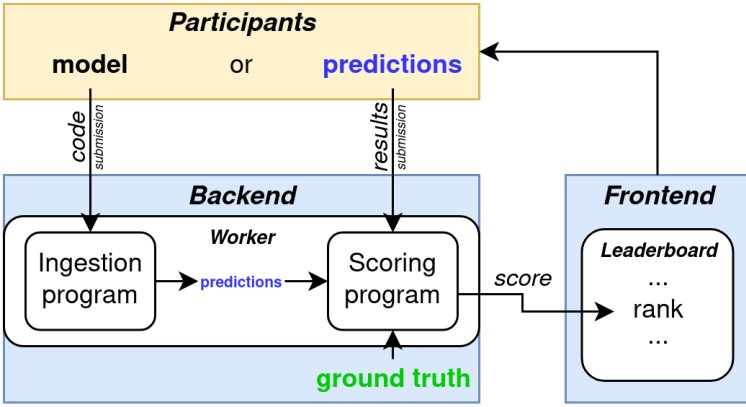

Figure 2: General competition workflow on CodaLab

**The starting kit**. If you have already joined a challenge as a competitor, you probably know how important it is to have a good starting kit. The goal of the starting kit is to provide participants with all the necessary resources to facilitate their dive into your competition, such as some example submissions, Jupyter notebooks, or any useful documentation and files. You can even provide the competition bundle itself (without the ground truth) inside the starting kit; this way, all the internal functioning will be perfectly transparent for the participants.

**Queues and Docker**

The public servers provide default compute workers. However, to run computationally demanding competitions, **organizers can create custom queues and attach their own CPU or GPU compute workers** (physical or virtual machines on any cloud service) to it. This modular architecture of *CodaLab Competitions* has been a key ingredient in growing its user base, without requiring that the institution hosting the main instance covers all computational costs. Another interesting aspect of this feature is that the training and testing of algorithms can be done on confidential data, without any leakage, by putting data directly inside the compute workers. This is especially useful for medical research, challenges organized by industries, and in other restricted domains.

The workflow of the jobs, showed in Figure 3, works as follows: each competition can be linked to only one queue, but the queue can be used by several competitions. Each worker can listen to only one queue, but the queue can be linked to several workers.

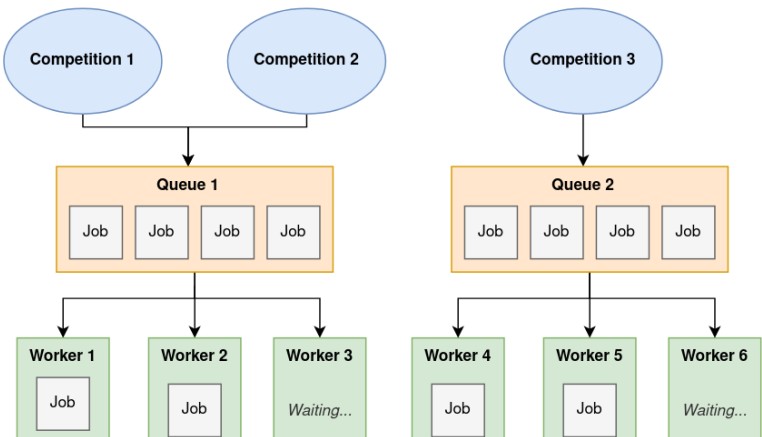

Figure 3: Diagram of the structure of workers and queues. The queues dispatch the jobs between the compute workers. Note that a queue can receive jobs (submissions) from several competitions, and can send them to several compute workers.

To setup a machine as a compute worker, you need to install Docker, note down the URL of your queue, and run a single command line[23].

The number of workers can be adjusted at anytime. This means that you can add more workers to the queue during the competition, they will dispatch all jobs automatically, increasing your computing power in real time in order to fit the needs of your competition.

One aspect that allows the same machine to compute jobs from many different competitions is the use of **Docker environments**. The execution of participants code and scoring are performed inside a container, which prevents to damage the servers and allows organizers to define a custom controlled environment for their competitions. Organizers can create a fully customized environment, with allowed libraries and programming languages for their participants' submissions, and simply link it to a competition by providing a DockerHub name and tag. This means that every candidate is judged in the same way, the competition does not get deprecated after some time and adding new libraries or updating the experimental environment is straightforward and transparent.

**Organizer features**

As a competition or benchmark creator, you have access to useful organizer-only features. These features are accessible from the grey buttons at the top of the user interface, as shown in Figure 4.

---

[2]https://github.com/codalab/codalab-competitions/wiki/User_
Using-your-own-compute-workers

[3]https://github.com/codalab/codabench/wiki/Compute-Worker-Management---Setup

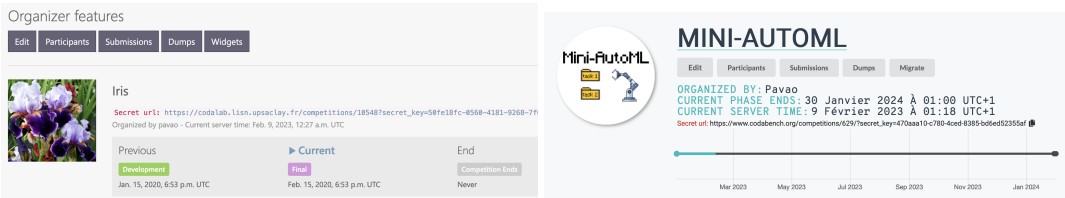

Figure 4: As an organizer, you have access to many interesting features, accessible from the grey buttons at the top of the interface of *CodaLab Competitions* (left) or *Codabench* (right).

**Editor.** The editor allows organizers to edit a competition that is already up and running. From this panel, you can edit every setting, and replace the data, scoring program or ingestion program.

**Manage participants.** You can choose to allow anybody to join your competition, or to have a registration process and validate who can join. You can accept or revoke access of the participants at any time.

**Manage submissions.** A panel to manage all the submissions made by the participants. From this panel, you can access details about the submissions: their date and author, the output and logs of the scoring programs and the submissions themselves (the files uploaded by the participants). The interface can be used to cancel, remove or re-run submissions. Overall, it is very useful to debug, to prevent cheating or to run post-challenge analysis.

**Dumps: export your competition.** A feature that you can use to download your competition as a bundle. All changes made directly through the editor will be saved and the resulting bundle can be re-uploaded on the platform or on any other instance of it. If you wish to re-upload the bundle on the same platform and want to keep the bundle as light as possible, you can use the option "use URI keys instead of files", in this case the datasets and programs will be referred by their address in the storage.

**Publish your competition.** By default, your competition is private. When a competition is private, it can be accessed only by its administrators, or by anyone from the "secret URL". Once you publish your competition, anyone can access it from the public URL. You can make your competition private again at any time.

**Auto-migration.** Automatically re-run the leaderboard's submissions from one phase to another phase.

## 3 CodaLab Competitions tutorial

### Get started

To create your first machine learning challenge, all you need to do is to upload a **competition bundle**. A competition bundle is a ZIP file containing all the pieces of your competition: the data, the documentation, the scoring program and the configuration settings, as explained in Section 2. Let's start from an example; it's the easiest way. Here is the competition bundle of the *Iris Challenge*, based on the famous Fisher's dataset Fisher (1936): Iris Competition Bundle. Now, go to CodaLab, and upload the file named "iris_competition_bundle.zip" as shown in Figure 5. Go to "*My Competitions*", then "*Competitions I'm Running*", and finally "*Create Competition*". During the last step you will be redirected to the final menu where you can upload the competition

bundle. After uploading it, you will see the message of Figure 6 indicating that your competition is successfully created and ready to receive submissions.

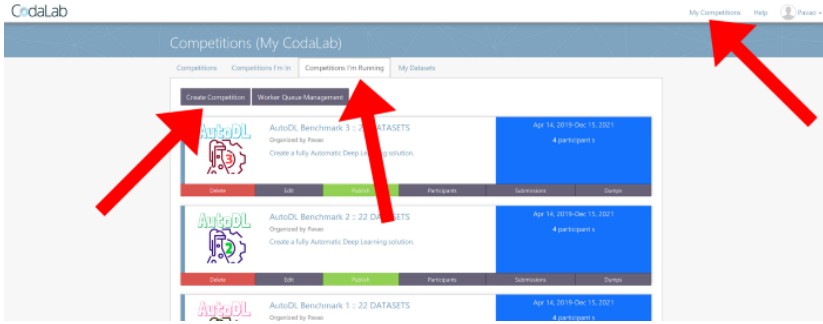

Figure 5: Go to "*My Competitions*", then "*Competitions I'm Running*" and finally "*Create Competition*".

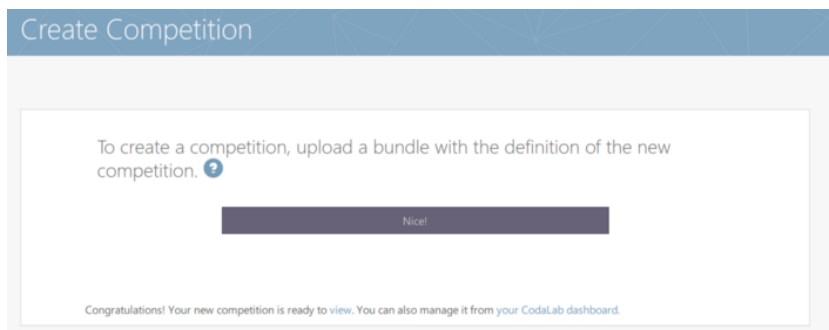

Figure 6: That's it! Your competition is ready to receive submissions.

Once the competition is uploaded, you can begin to make submissions. To do so, UnZip the downloaded bundle, go inside the starting kit and zip the content of either the "sample_result_submission" or the "sample_code_submission" folder. It is important to zip the files without directory structure and to include the "metadata" file in the case of code submission. Then, on the website (see Figure 7), go to the "*Participate*" tab, then "*Submit / View results*", click on "*Submit*" and select your zip file. The submission will process for a few moments. After that, you'll be able to see your score in the leaderboard, in the "*Results*" tab.

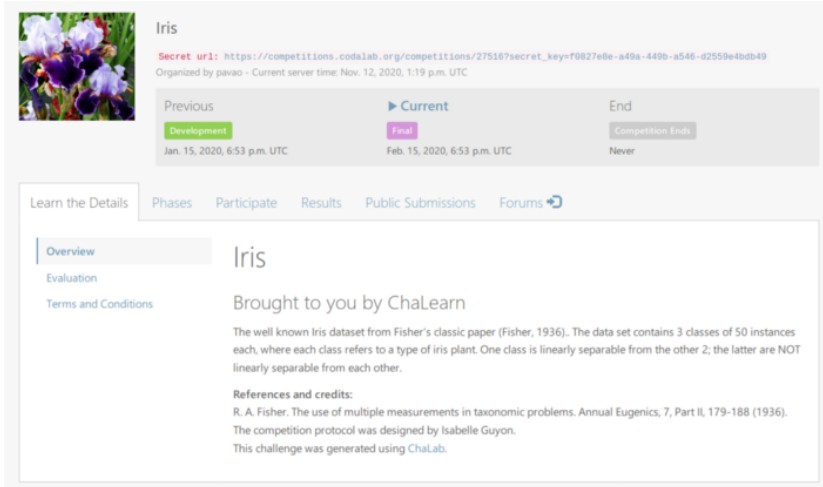

Figure 7: Main page of the Iris Challenge. The web pages are defined by the HTML files from the bundle.

**Customize competition**

Your competition is up and running! However, you wish to edit it. It is still possible. As an administrator of your own challenge, you have access to the "Edit" menu: a panel in which you can edit every setting. Here are some examples:

- Force submission to leaderboard: if enabled, the last submission of a participant is the one showed on the leaderboard.

- Disallow leaderboard modifying: if disabled, users can select which of their submissions appear on the leaderboard.

- Share administrator rights: you can add *CodaLab* users as administrators of your competition by giving their username or email address. They'll have access to all organizer-only features, except deleting the competition.

- Anonymous leaderboard: if enabled, the username are hidden in the leaderboard.

- Registration required: if enabled, users need to request the access to your competition. You then have to accept or reject their participation manually from the "Participants" tab.

- Select the queue.

- Specify competition docker image by its DockerHub name.

- For each phase you can chose different data and scoring programs.

If you wish to change the dataset or the scoring program, you'll first need to upload the new version from the "*My Datasets*" page, as shown in Figure 8. You will then be able to select it in the editor.

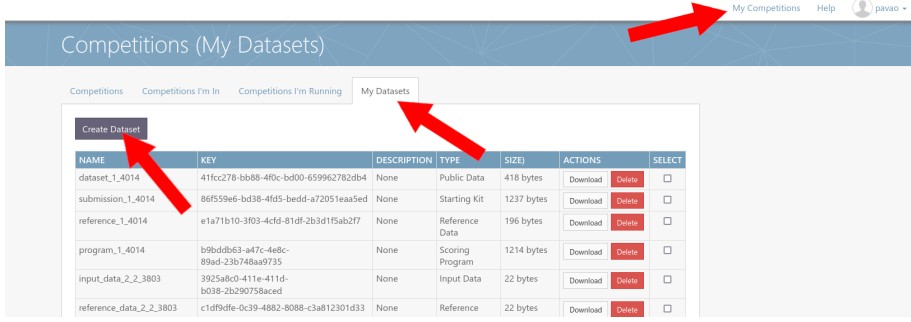

Figure 8: Go to "*My Datasets*" to upload new datasets or programs.

## 4 Codabench tutorial

### Get started

As an evolution of *CodaLab Competitions*, *Codabench* is similar in terms of functioning and features. In this part of the tutorial, we will organize a simple benchmark on *Codabench*, in which the participants can have multiple entries displayed on the leaderboard. Note that competition bundles from *CodaLab Competitions* are compatible with *Codabench*, while not vice versa.

Let's have a look at the Mini-AutoML Bundle. To upload the file named "bundle.zip" to *Codabench*, go to "*Benchmarks*", then "*Management*" and finally "*Upload*" (Figure 9). Then click on the paper clip button to select and upload the bundle (Figure 10).

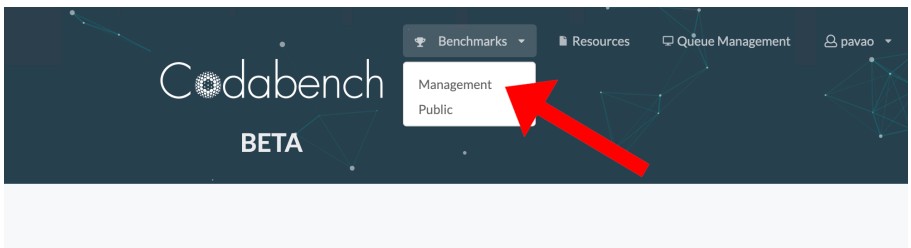

Figure 9: Go to "*Benchmarks*", then "*Management*" and finally "*Upload*".

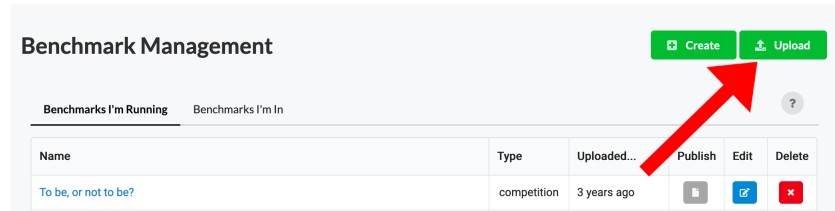

Figure 10: Then click on the paper clip button to select and upload the bundle.

The main page of the benchmark should look like Figure 11.

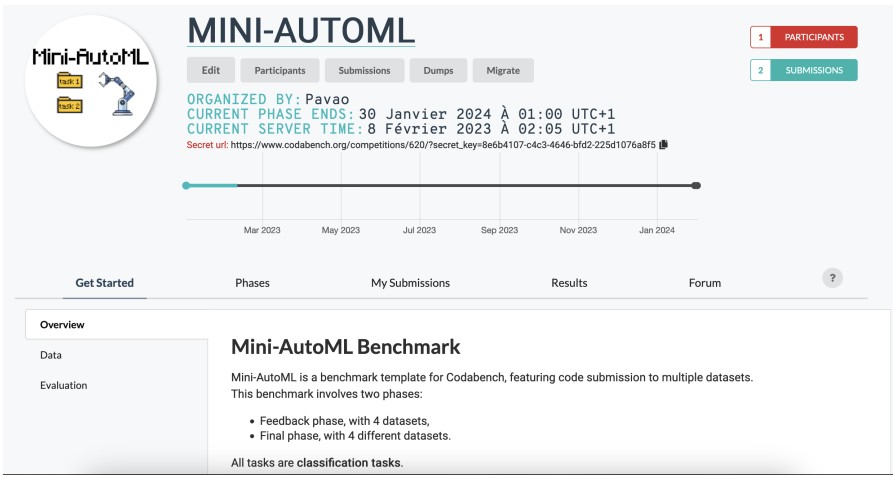

Figure 11: Overview of Mini-AutoML main page.

Let's now make a submission. To do so, download the file sample_code_submission.zip and upload it in the "*My Submissions*" tab of the benchmark. You will be able to view logs in real time during the processing of the submission. The leaderboard, available in the "*Results*" tab of the benchmark, will then be updated.

**Customize benchmark**

Let's customize this newly created benchmark, with the goal in mind to be able to fill in the leaderboard with many different models, in order to compare them.

**Fact sheets.** In the first tab of the editor, you can enable "fact sheets" to gather more information about the submissions. Enabling fact sheets means that the participants will be asked to fill in some information when making submissions. You can fully customize the information fields, making them required or not, and choosing which information appears on the leaderboard. This can be used to display the name of the methods, the URL to the source code, or a description of the method. The gathering of metadata about the methods used is crucial when conducting a benchmark, and this interface makes it simple to gather all this information in one place.

**Edit data and programs.** *Codabench* provides an interface to upload *Ingestion Program*, *Scoring Program*, *Starting Kit* and *Data (public data, input data and reference data)*. To upload a new dataset or program, go to "*Resources*", "*Datasets*" and click on "*Add Dataset*". Then name it, select your ZIP file, and chose the type (reference data, scoring program, etc.) (Figure 12).

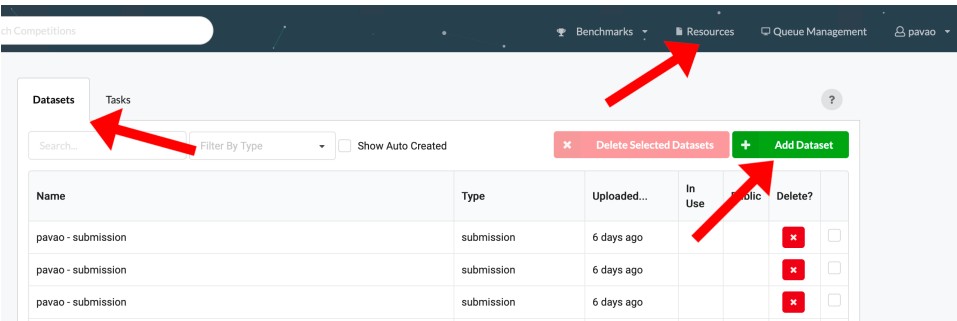

Figure 12: To upload a new dataset or program, go to "*Resources*", "*Datasets*" and click on "*Add Dataset*". Then name it, select your ZIP file, and chose the type (reference data, scoring program, etc.)

Once these are uploaded, a task can be created using the uploaded files. A task is a combination of *Ingestion Program*, *Scoring Program*, *Input Data*, and *Reference Data*. o create a task, go to "*Resources*", "*Tasks*" and click on "*Create Task*". Then name it and select the files previously uploaded: Input data, reference data, ingestion program and/or scoring program (Figure 13).

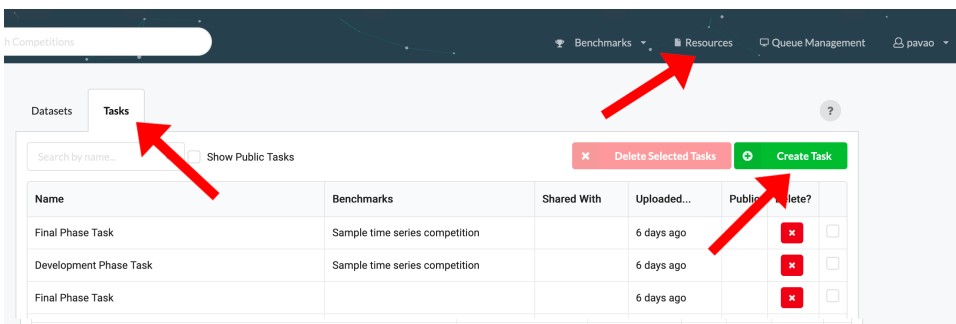

Figure 13: To create a task, go to "*Resources*", "*Tasks*" and click on "*Create Task*". Then name it and select the files previously uploaded: Input data, reference data, ingestion program and/or scoring program.

In your competition editor, you can add, update or delete a task for each phase. Unlike *CodaLab*, *Codabench* allows you to have multiple tasks for each phase. To associate a task to a phase, go to the editor, then "*Phases*", click on the edit button and select the desired task (Figures 14 and 15).

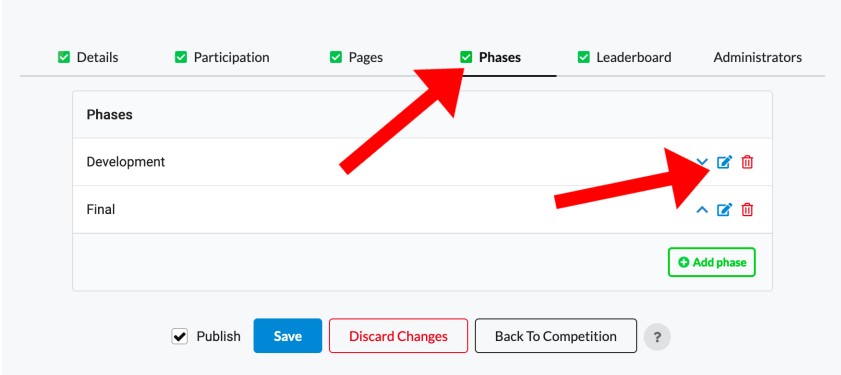

Figure 14: Go to the editor, then "*Phases*" and click on the edit button.

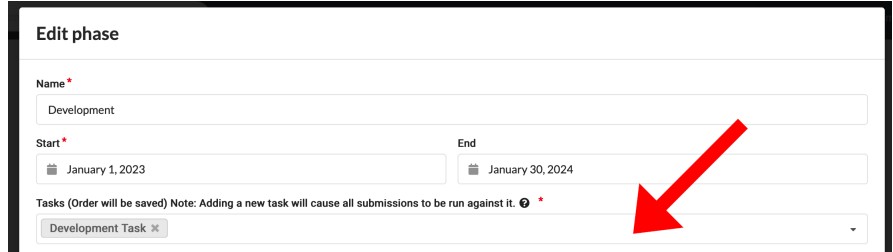

Figure 15: You can now associate the desired task to the phase.

**Submission rules.** The submission rule programs the behavior of the leaderboard regarding new submissions. Submissions can be forced to the leaderboard or manually selected, can be unique or multiple on the leaderboard, etc. To edit the submission rule, go to the editor, then "Leaderboard" and edit the leaderboard (Figure 16). Then change the submission rule from "Force Last" to "Add And Delete Multiple" (Figure 17). "Force Last" means that only the last submission of each participant will be shown on the leaderboard, while "Add And Delete Multiple" means that the participants will be allowed to manually select multiple submissions to show on the leaderboard.

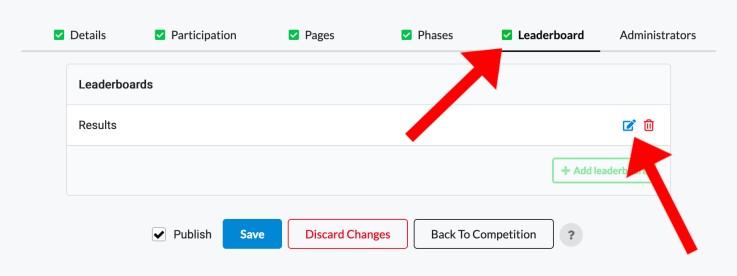

Figure 16: To edit the submission rule, go to the editor, then "Leaderboard" and edit the leaderboard.

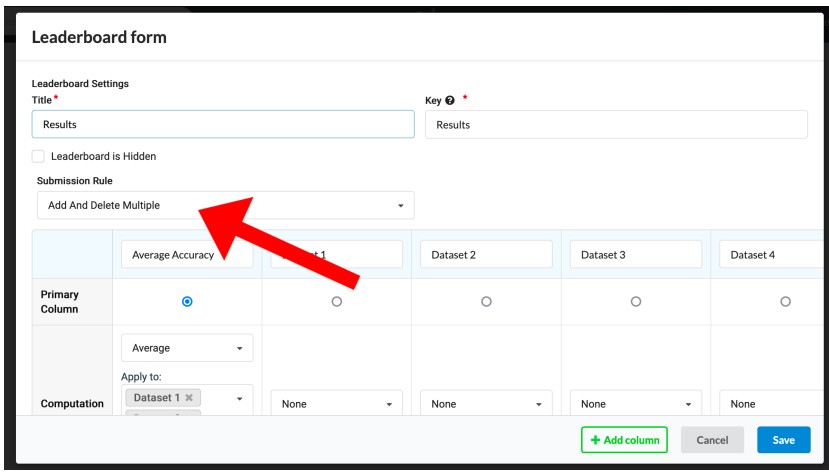

Figure 17: Change the submission rule from "Force Last" to "Add And Delete Multiple".

The current submission rule, "Force Last", means that only the last submission of each participant will appear on the leaderboard. This is a classical setting for competition. Changing this rule to "Add And Delete Multiple" will allow the participants to manually select which submissions will appear on the leaderboard, and multiple submissions per participant on the leaderboard are allowed.

**Add submissions to the leaderboard.** Now that the leaderboard is set up, let's submit different variations of the code of the model from the sample_code_submission.zip. This example code submission simply calls a classifier from Scikit-Learn Pedregosa et al. (2011). Replace the *DecisionTreeClassifier* with the classifier of your choice. Remember to differentiate the different submissions by filling the "Method name" in the fact sheet. Once your submission is processed, click on the leaderboard button under "Actions" in the submissions table to manually add them to the leaderboard (Figure 18).

| ID # ▼ | File name | Date | Status | Actions |
|---|---|---|---|---|
| 4401 | knn.zip | 2023-02-09 02:16 | Finished | ▦ ⦕ |
| 4400 | mlp.zip | 2023-02-09 02:16 | Finished | ▦ ⦕ |
| 4399 | gaussiannb.zip | 2023-02-09 02:16 | Finis | ▦ ⦕ |
| 4398 | rf.zip | 2023-02-09 02:14 | Finished | ▦ ⦕ |
| 4397 | rf.zip | 2023-02-09 02:14 | Finished | ▦ ⦕ |
| 4395 | sample_code_submission.zip | 2023-02-08 16:30 | Finished | ✓ ⦕ |

Figure 18: Once your submission is processed, click on the leaderboard button under "*Actions*" in the submissions table to manually add them to the leaderboard.

The leaderboard finally looks like Figure 19.

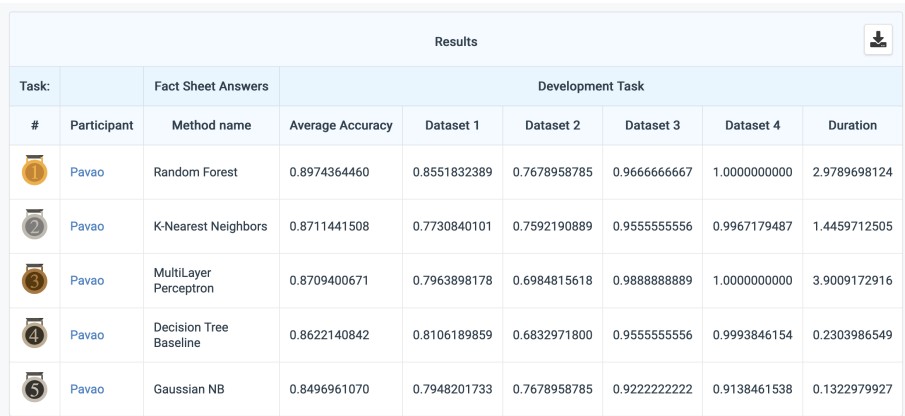

Figure 19: Screenshot of the filled up leaderboard. The random forest classifier did the best job in the first phase of the benchmark!

## 5 Conclusion

Congratulations! You have learned the basics of *CodaLab Competitions* and *Codabench*, and now you can organize your own competitions or benchmarks! However, we barely scratched the surface of all the possibilities offered by these platforms. To learn more, you can refer to: CodaLab Competitions Documentation and *Codabench* Documentation.

From the documentation, you will learn how to link your personal compute workers (CPU, GPU), how to customize the ingestion and scoring programs, how to define complex leaderboards with multiple criteria, or even how to deploy your own instance of the platform. You can also join the effort and develop your own features!

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

# Special designs and competition protocols

**Wei-Wei Tu**                                                    TUWEIWEI@4PARADIGM.COM
*The 4th Paradigm*
*China*

**Adrien Pavão**                                              ADRIEN.PAVAO@GMAIL.COM
*LISN, CNRS*
*Université Paris-Saclay*
*France*

**Reviewed on OpenReview:** *https://openreview.net/forum?id=aATSw1zwOV*

## Abstract

With the development of AI technology, many novel machine learning frameworks have been raised and applied in AI academic and industry research and business application. Organizing competitions in these areas can greatly help the research and development of related algorithms and technology. In this chapter, we explore the design of competitions in various kinds of machine learning field: supervised learning, automated machine learning, metalearning, time series analysis, reinforcement learning, adversarial learning, and using confidential data. For each of these specific competition protocol, we discuss the framework and design of the competition process. We believe this chapter can make great help to both the organizers and the participants, therefore accelerate the development of AI industry and research.

**Keywords:** competition, design, supervised learning, automated machine learning, metalearning, time series analysis, reinforcement learning, adversarial learning, confidential data

## 1 Introduction

Machine learning is an expansive field offering a rich diversity of algorithms, each developed to solve specific tasks. These algorithms are commonly grouped into three main categories: supervised learning, unsupervised learning, and reinforcement learning algorithms. Beyond the algorithms themselves, the possibilities are further augmented by the diversity of data and domains of applications. Depending on the nature of the data, its source, shape, quantity and patterns, different approaches are required. The applications of machine learning are virtually limitless, covering medicine, physics, natural language processing, economics, and more. To be able to capture this complexity and diversity in competitions and benchmarks, innovative experimental design is required.

In this chapter, we analyse the features and special designs about the challenges and benchmarks of supervised learning (Section 2), time series analysis (Section 2.1), automated machine learning (Section 3), metalearning (Section 3.1), reinforcement learning (Section 4), adversarial learning (Section 5), and using confidential data (Section 6). We also give some tips about how to perform well in these competitions as participants.

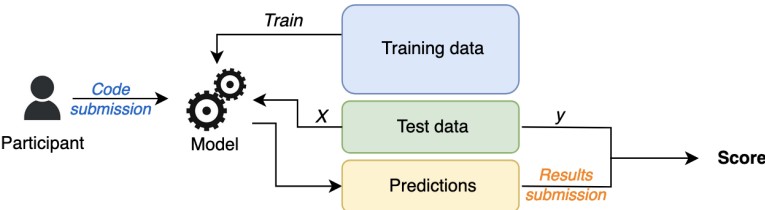

Figure 1: Supervised learning evaluation workflow. Models are trained and subsequently evaluated on the withheld test set. The two possible protocols, *code submission* and *results submission*, are illustrated. *X* represents the features, and *y* the ground truth, of the test set.

## 2 Supervised learning

Supervised learning is a foundational paradigm in machine learning where models are trained using labeled data. In this paradigm, for each input instance in the dataset, there is an associated correct output, commonly referred to as *label* or *ground truth*. The primary goal of supervised learning is to construct a model capable of making accurate predictions for unseen instances based on this training.

In a classic supervised learning competition, participants evaluate their models on a given task using a dataset split into training and test sets. Typically, as depicted in Figure 1, participants are provided with a training dataset to develop their models, and the evaluation is conducted on the withheld test set. Each competition phase must feature a different test set to prevent overfitting. Importantly, participants should not have access to the labels of the test set.

The choice of evaluation metrics in supervised learning challenges typically depends on the nature of the task — be it regression, classification, or others. The underlying goal is to objectively measure the performance of submitted models in terms of *accuracy*, *precision*, or other relevant metrics. Further insights into evaluation metrics are provided in the chapter 4.

### 2.1 Time series analysis

Time series analysis includes a wide variety of tasks, such as anomaly detection, sequence-to-sequence problems, or survival analysis, each presenting unique specificity. In the this section, our discussion centers around two central time series tasks: *time series regression* and *time series forecasting* (or prediction). While time series regression (Section 2.1.1) involves modeling the relationship between a dependent time-indexed variable and one or more independent variables, aiming to understand or predict the dependent variable's variations over time, time series forecasting (Section 2.1.2), on the other hand, is primarily concerned with predicting future values of a series based on its own past values and inherent patterns. This distinction is highlighted by Figure 2. The key distinction lies in the fact that regression models are more general and can be applied to predict values at any point in time, not strictly in the future, whereas forecasting is explicitly future-oriented, leveraging the temporal order of data to make predictions. Non sequential meta-data may also be available.

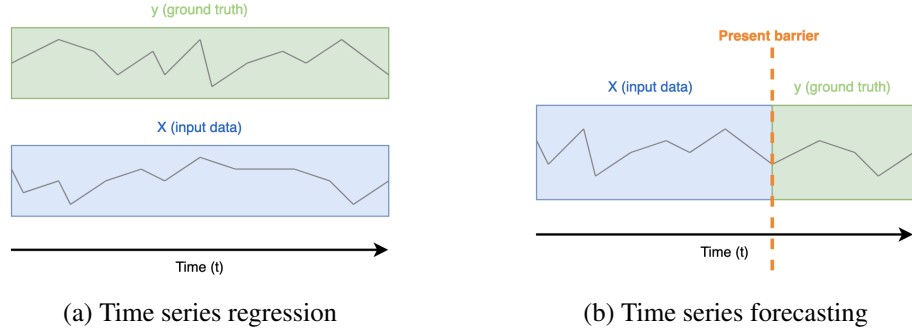

(a) Time series regression

(b) Time series forecasting

Figure 2: Schematic view of time series regression (left) and time series forecasting (right). In regression, predictors can use past, present, or future values, while in forecasting, the task is to predict future values based on historical data and trends.

### 2.1.1 TIME SERIES REGRESSION

Time series regression is essentially a supervised learning task with a temporal dimension, where the goal is to predict a continuous target variable based on historical data. While it shares similarities with classical regression in terms of learning from input-output pairs and minimizing prediction error, the time component introduces dependencies between observations, necessitating consideration of the order and timing of data points in the modeling process. Time series regression can be multivariate, meaning that multiple variables must be predicted, as it is the case in the *Paris Region AI Challenge 2020* (PRAIC) (Pavao et al., 2021). In this case, the performance can be measured using an average score (weighted or not) across the output variables, or using any ranking function. Another example of time series regression competition is the *AutoSeries Challenge* (Xu et al., 2021), which happens to be also an AutoML competition. This competition confirmed the efficiency of Gradient-Boosting Machines (GMB) to tackle time series regression tasks, as well as random search hyper-parameter tuning to tackle the AutoML part of the problem.

### 2.1.2 TIME SERIES FORECASTING

"*A Brief History of Time Series Forecasting Competitions*" by Hyndman (2023) traces the transformative impact of forecasting competitions from the *Makridakis Competitions* series, organized by Spyros Makridakis and spanning from 1980 to today (Makridakis et al., 1982; Makridakis and Hibon, 2000; Makridakis et al., 2018), highlighting their role in shaping forecasting methodologies across diverse data types. The paper emphasizes the consistent success of combination forecasts, encouraging the use of ensemble methods, and points out the balance needed between automated forecasting and domain-specific expertise. Other exemplary time series prediction competitions include the competitions organized at the Santa Fe Institute (Weigend and Gershenfeld, 1993) which contributed to our understanding of time series prediction in a variety of contexts.

Time series forecasting competitions can be designed in both interactive or non-interactive settings, depending on the objectives and constraints of the challenge. In a **non-interactive** format, participants are provided with a complete dataset up to a certain point in time, and they are required to make predictions for future data points. The models are then evaluated based on their accuracy in predicting these unseen data points. This format is straightforward but may not fully capture the dynamic nature of real-world time series forecasting, where new data continuously become available,

and models need to be updated accordingly. On the other hand, **interactive competitions** aim to mimic these real-world conditions by releasing data in stages. Participants make predictions based on available data, and as the competition progresses, new data are released, which can be used to update and improve the models. This format encourages the development of adaptive models that can respond to changes in data patterns over time. This design was typically found in the *COVID-19 Global Forecasting*[1] challenge on Kaggle, where participants were tasked with predicting the spread of COVID-19 disease. Subsequently, the initially unknown ground truth was revealed and added to the training data on a weekly basis.

## 3 Automated machine learning

Automated machine learning (AutoML) is a field of study that focuses on developing methods and systems that can automate the process of building machine learning models. The goal of AutoML is to make it easier to build accurate and effective machine learning models without requiring extensive human intervention. By nature, AutoML methods are built to be able to solve a wide variety of tasks. Examples of such competitions include the AutoML Challenge Series (Guyon et al., 2019), the AutoDL Challenge Series (Liu et al., 2021), the AutoML Decathlon (Roberts et al., 2022) and the AutoML Cup (Roberts et al., 2023) based on the NAS-Bench-360 benchmark (Tu et al., 2023).

The general competition protocol consists in evaluating the candidate algorithms on a set of $m$ tasks. For each of these tasks, the model is trained from scratch and evaluated on a hold-out test set, as demonstrated by the diagram in Figure 3. The $m$ scores that result from evaluating the algorithm across the tasks are subsequently fed into to a master scoring and ranking process. Although the scores obtained on various tasks could simply be averaged, we suggest computing the average of the ranks achieved by comparing all candidates across the given tasks. This approach ensures a more robust ranking that accurately reflects the aim of a competition in automated machine learning. This point is explained in details in the chapter 4.

A key aspect of the experimental design of automated machine learning competitions and benchmarks is the **blind testing**. To accurately assess a model's capability to solve diverse and unrelated tasks, participants must not have access to the test data. While some example training datasets can be made available to help participants in developing their models, the feedback and final evaluation stages must be conducted blindly, typically through the submission of code rather than direct interaction with the test data.

The selection of datasets and evaluation metrics is flexible and intrinsically tied to the specific objectives of the challenge. The main principle is that greater diversity in datasets is likely to yield a winning solution with more general applicability. Reciprocally, using similar datasets and metrics is more likely to produce an algorithm specialized in a particular domain or task. Having a large number of different tasks, while computationally expensive, enhances the overall diversity of the model's capabilities. This topic is discussed in the chapter 4.

While most AutoML challenges focus on supervised learning tasks, classification and regression, this experimental design can be used to organize crowd-sourced competitions or benchmarks on the automation of other machine learning tasks, such as data processing, clustering, content recommendation and more. The pre-requisite is to have a scoring metric defining the objective of the problem.

---

[1] https://www.kaggle.com/c/covid19-global-forecasting-week-1

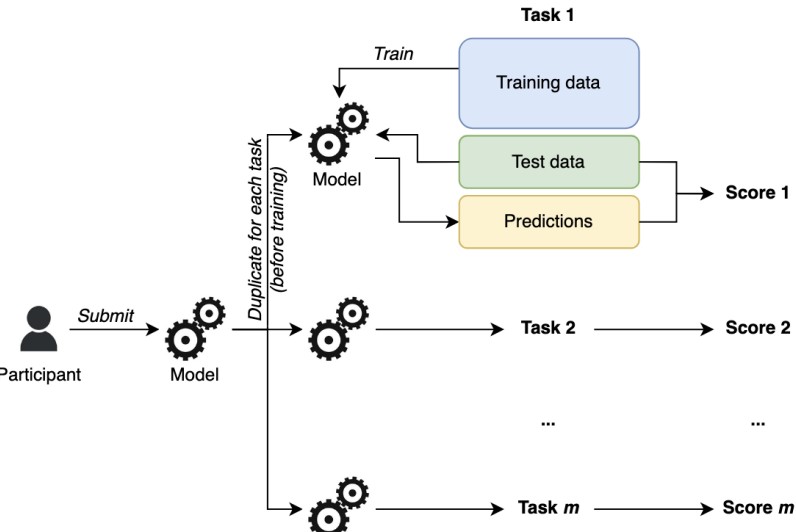

Figure 3: Automated machine learning evaluation workflow. The submitted model is trained and tested from scratch on a set of independent tasks.

## 3.1 Meta-learning

Meta-learning is a sub-problem of AutoML. In its general definition, AutoML is a process of automating the machine learning process, including tasks such as data preprocessing, feature engineering, model selection, and hyperparameter tuning. AutoML techniques use algorithms to search for the best machine learning pipeline automatically. On the other hand, meta-learning is focused on learning how to learn. Meta-learning algorithms learn from experience to adapt their learning strategies for different tasks and domains (Brazdil et al., 2022). In essence, while AutoML automates the process of finding the best machine learning pipeline for a specific task, meta-learning takes a step further and automates the process of improving the learning algorithm's generalization capability across multiple tasks.

In the meta-learning challenge protocol proposed by El Baz et al. (2021); Baz et al. (2021) for the Cross-domain MetaDL Challenge, the evaluation of the candidate algorithms is divided in two sequential phases: the meta-training and the meta-test. During the meta-training, the submitted algorithm is trained on a set of datasets. The trained model is then forwarded to the meta-test, where it will be trained and evaluated separately on a new set of tasks. The whole process is illustrated by the diagram in Figure 4. The set of scores produced is then used to compare the model with other candidate models. As for the AutoML Challenge, the entire process is conducted blindly, preventing the participants from adapting their approaches to the specific datasets used. The main difference with the AutoML protocol (Section 3) is the use of a controlled meta-training phase, which implies that all candidate algorithms are pre-trained on the same data.

## 4 Reinforcement learning

Reinforcement Learning (RL) is a subset of machine learning where an agent learns to make decisions by interacting with an environment. The agent receives feedback in the form of rewards

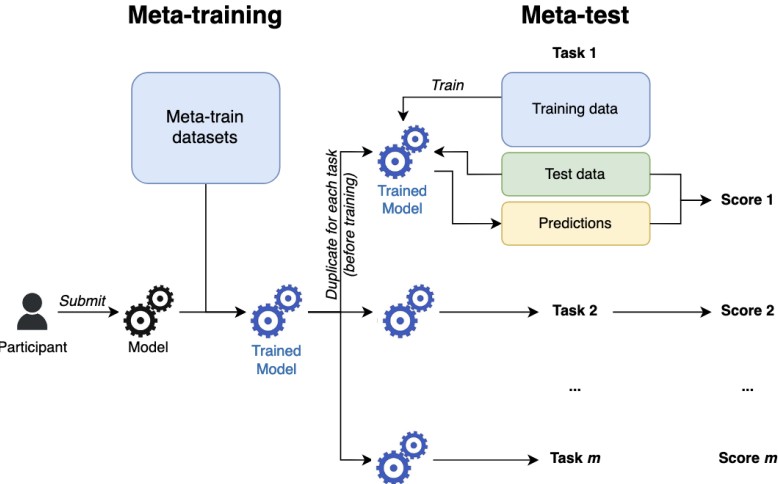

Figure 4: Meta-learning evaluation workflow. The submitted model is trained on meta-train datasets, and then it is tested on a set of meta-test tasks.

or penalties, guiding it to optimize its behavior to maximize cumulative rewards over time, as illustrated by Figure 5. RL has been successfully applied in various domains, including robotics, game playing, and autonomous vehicles. Organizers of such challenges must choose suitable problem simulations, balance environmental complexity with computational demands, and set objective evaluation criteria that consider efficiency, adaptability, and robustness of the agent's performance.

Designing challenges for RL is an inherently complex task. One of the primary difficulties is the requirement for a simulated or real-world environment where participants' algorithms can interact, learn, and be evaluated. Ensuring the stability, reliability, and realism of these environments is crucial, as inconsistencies or inaccuracies can lead to misleading results and prevent the learning process. Furthermore, RL algorithms typically require a substantial amount of interactions with the environment to learn effectively, making the computational cost a significant consideration. Additionally, there is no one-size-fits-all metric for assessing the performance of RL algorithms across various tasks and environments, necessitating the careful selection and design of evaluation criteria that accurately reflect the objectives of the specific competition.

RL challenges can be designed following different protocols, primarily distinguished by the availability of pre-collected data. In challenges **without** pre-collected data, the algorithms proposed by participants engage directly with the environment, allowing data acquisition and learning concurrently. On the other hand, challenges **with** pre-collected data enable participants to refine and train their algorithms in an offline manner. The setting without pre-collected data is often referred to as **online learning**, and typically occurs in *OpenAI Gym Competitions* such as the *Retro Contest* (Nichol et al., 2018) where agents interacts with video games environment without prior knowledge. This approached is opposed to **offline learning**, which uses pre-collected data, such as the Atari Grand Challenge dataset (Bellemare et al., 2013). This particular dataset comprises a collection of human demonstrations across various of Atari games. In offline learning scenarios, the algorithm learns exclusively from this existing data, without the opportunity for real-time interaction or data acquisition, as seen in online learning settings. Regardless of whether algorithms are

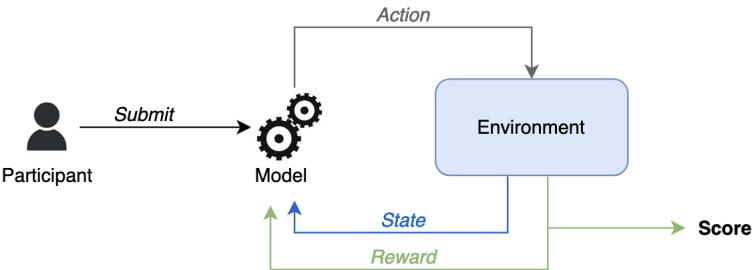

Figure 5: Reinforcement learning competition workflow. The submitted agent interacts with the environment, with the objective of maximizing the reward over time.

trained through online or offline learning protocols, their performance must ultimately be evaluated by interacting to live environments.

It is common to use a cumulative reward over time as a primary metric for determining the final score of participants' models. In such settings, the manner in which time is quantified plays a crucial role in the evaluation process, introducing a potential challenge in ensuring equitable and unbiased benchmarking. To mitigate inconsistencies that may arise from hardware disparities, it is advisable to standardize the measurement of total time in terms of the **number of environmental steps** taken, rather than relying on real-time duration measured in seconds. Adopting this approach ensures a consistent and equitable evaluation framework, as it remains invariant across different hardware configurations, thereby enhancing the fairness and reliability of the competition results.

Moreover, in RL challenges, having well-defined and expert-crafted metrics is crucial for evaluating the performance of participating algorithms accurately and fairly. These metrics need to capture not just the immediate rewards but also the long-term impact of decisions made by the RL agents. The design of these metrics requires a deep understanding of the specific domain, the goals of the RL task, and the potential trade-offs between different objectives.

Another interesting distinctions in RL competition design is **interacting with the environment** versus **interacting with other agents**. In the "interaction with environment" setting, the agents submitted by participants are evaluated by interact with the given environment. The related scenes includes single-player games, auto-driving, robot controlling, etc. In the "interaction between agents" setting, the agents are ranked by the performance of competition with other agents. The related scenes includes for instances multi-player games and stock market.

Figure 6 shows the process about the interacting with environment setting and the Figure 7 shows the process about the interacting with other agents setting reinforcement learning competition. The most important issue in the design of reinforcement learning competition is how to evaluate the ranking of submissions. In the setting of interacting with environment, the performance of submissions can be measured by the cumulative reward the submission gained by interacting with the environment. So it is a very important challenge for competition organizers to construct a good simulating environment. The quality of environment determines the quality of the whole competition. In the setting of interacting with other agents, the most popular way is to model the ranking score by a Gaussian distribution $N(\mu, \sigma^2)$. Submissions with similar skill rating would be picked to finish a match. The winner's $\mu$ will be increased while the loser's $\mu$ will be decreased. If there

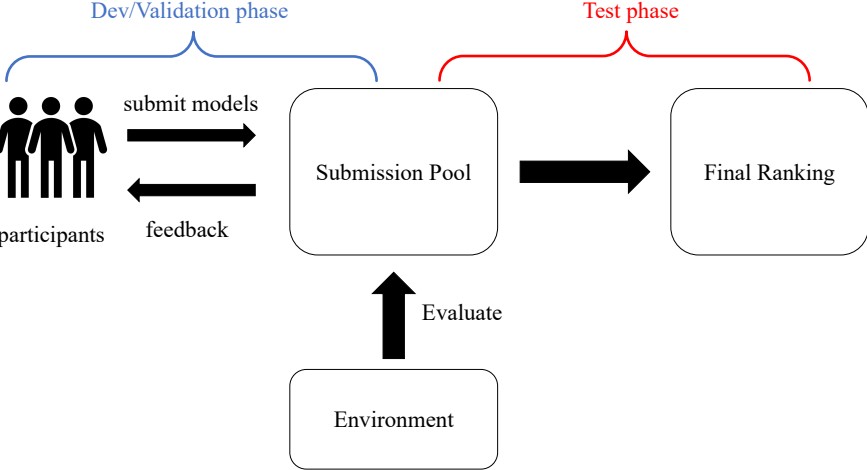

Figure 6: Process of reinforcement learning competitions of interacting with environment.

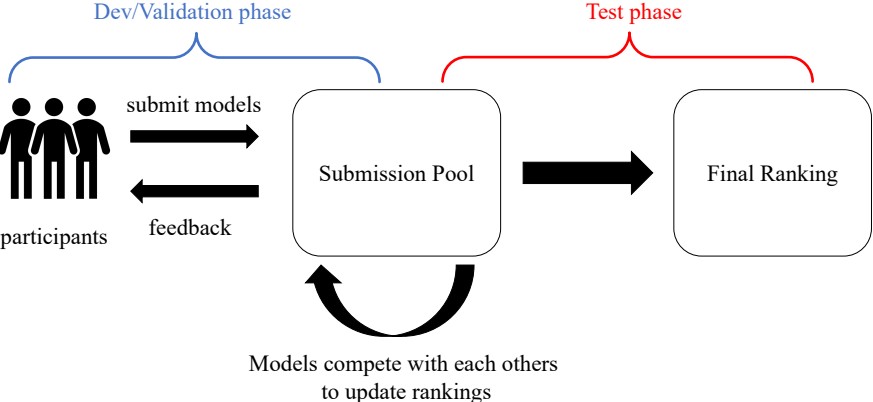

Figure 7: Process of reinforcement learning competitions of interacting with other agents.

is a draw, the $\mu$ of two teams will be moved closer to their mean. Here the key issue is how pick submissions with similar ranking score.

Instances of RL competitions include those in the domain of biomechanics. Notable examples in this category are the *Learning to Run* challenge (Kidzinski et al., 2018) and the *AI for Prosthetics* competition (Łukasz Kidziński et al., 2018). These events enabled progress in simulating and understanding complex biomechanical processes. In addition to biomechanics, RL competitions also extend to video game environments, offering unique challenges that require agents to navigate and interact within virtual worlds. The *MineRL Competition* (Guss et al., 2019) and the *Procgen Competition* (Mohanty et al., 2021) typically test the ability of RL algorithms to adapt and perform across procedurally generated environments. Furthermore, competitions such as *Metalearning from Learning Curves* (Nguyen et al., 2022) explore general aspects of machine learning. This challenge study the meta-analysis of learning processes, encouraging the development of algorithms that can learn effectively from existing learning trajectories.

Two notably interesting examples of past reinforcement learning competitions are the *NetHack 2021 NeurIPS Challenge* and the *Google Research Football with Manchester City F.C.*.

Held by Meta and DeepMind, the **NetHack 2021 NeurIPS Challenge** required participants to design an agent to play the game NetHack automatically. NetHack is a single-player video game in which the player is required to navigate the procedurally generated, ascii dungeons to find the amulet. Although it is a very complex game, it can be simulated efficiently by the NetHack Learning Environment (NLE) which is presented at NeurIPS 2020. This competition was split into a development and test phase. During the development phase, participants were able to submit their agents to the leaderboard once a day and 512 evaluation runs would be performed to calculate a preliminary place on the dev-phase leaderboard. The top 15 participants for each track were taken from the dev-phase leaderboard and invited to join the test phase. In test phase, participants were able to submit their best agents 3 to the test-phase leaderboard and 4096 evaluation runs would be performed to calculate the final ranking Net (2020). 42 teams joined this competition and submitted 632 submissions to compete a total prize of 20,000 dollars.

The **Google Research Football with Manchester City F.C** competition was held on Kaggle in 2019 by Google Research and Manchester City F.C. foo (2020). In this competition, each team was required to create AI agents to control a 11-player football team, and them compete with other teams in the simulation environment Kurach et al. (2020). To simplify this challenge, at each time step, the team only need to control one player by choosing an action from a given set of 19 actions. Each submission had an estimated skill rating modeled by a Gaussian distribution $N(\mu, \sigma^2)$, and the rating was updated by the procedure mentioned above. 1,138 teams participated this competitions to compete a total prize of 6,000 dollars.

There are some tips for participants who wants to get good performance in reinforcement learning competitions. The first tip is to focus on the design of reward function, especially in some scenes the reward function is very sparse. The second tip is to design the feature processing model and reinforcement model structure carefully, because they are the key issues to speed up the training progress and improve the model performance. The third tip is to combine with some typical reinforcement learning algorithms such as MCTS and on-policy algorithms.

In conclusion, designing challenges for RL competitions is a nuanced task that requires careful consideration of the learning environment, computational resources, and evaluation metrics.

## 5 Adversarial learning

Recent research shows that many machine learning classifiers, especially deep learning models, are highly vulnerable to adversarial examples Biggio et al. (2013); Szegedy et al. (2014). An adversarial example is a sample of input data which has been slightly modified to mislead the classifiers while human observers can not notice the modification at all. The existence of adversarial samples raises a huge challenge to the security of machine learning and AI systems. Adversarial learning competition is an important way to examine the adversarial attack and defense algorithms, thus plays an important role in adversarial learning researches.

The adversarial learning competitions includes two aspects: attack and defense. In the attack competition, participants are required to attack a given model. There are three types of attack settings categorized by the revealed information of victim model.

- **White-box attack setting**. In this setting, participants have the full information about the victim model, including the model structure, the value of parameters and hyper-parameters.

- **Black-box attack setting**. In this setting, participants can not directly know the details about the victim model. Instead, participants can query the victim model certain inputs and observe the prediction results.

- **Universal attack setting**. In this setting, participants can not know anything about the victim model or query the victim model. It requires to construct the universal adversarial samples which can mislead most machine learning classifiers.

The attack setting can also be divided into targeted attack and non-targeted attack. In non-targeted attack the adversary only need to cheat the victim model to give a wrong prediction, while in the target attack the adversary is required to mislead the victim model to output a special given label as prediction.

In the defense competition, the participants are required to submit classifiers trained on the given dataset. Then the accuracy of submissions are measured on the adversarial samples constructed by certain adversarial attack algorithm.

Similar with the reinforcement learning competition design, one of the most important issue in adversarial learning competition design is how to evaluate and rank the performance of submissions. For the evasion attack, there are two dimensions in the measure about the attack performance: disturb norm and the attack success rate. For simplicity, we only discuss the case of non-targeted attack, the measure of targeted attack can be designed with similar method. The performance of submissions can be evaluated by the attack success rate with the restricted disturb norm. For example, the score of submitted attack model on test sample $x$ can be represented by

$$\text{score}(g,x) = \begin{cases} 1, \text{ if } ||x - g(x)|| \leq \varepsilon \text{ and } f(x) \neq f(g(x)); \\ 0, \text{ otherwise} \end{cases}. \tag{1}$$

Here $g$ represents the submission, $g(x)$ represents the adversarial sample produced by the submission model, $f$ represents the victim model, and the $\varepsilon$ represents the threshold of disturb norm. The performance of submissions can also be evaluated by the disturb norm of adversarial samples which attack the victim model successfully. The loss can be designed as follows:

$$\ell(g,x) = \begin{cases} ||x - g(x)||, \text{ if } f(x) \neq f(g(x)); \\ A, \text{ if } f(x) = f(g(x)) \end{cases}, \tag{2}$$

in which $A$ represents the penalty for adversarial samples failed to mislead the victim model. $A$ must be larger than all the possible disturb norms, i.e. $A \geq \max_{x,x'} ||x - x'||$.

In the evaluation of defense model, the influence of disturb norm should be considered, too. Similar with the evasion attack case, there are two ways to measure the performance of defense model. The first way is to test the defense models with adversarial samples with restricted disturbed norm, then compare the prediction accuracy of defense models on adversarial samples. The second way is to evaluate by the disturb norm of adversarial samples the submissions can distinguish successfully. For example, the score function of the second evaluation method can be designed as follows:

$$\text{score}(f,x') = \begin{cases} 0, \text{if } f(x') \neq y; \\ ||x - x'||, \text{if } f(x') = y \end{cases}, \tag{3}$$

in which $(x,y)$ represents the initial samples and the ground-truth label, and the $x'$ represents the adversarial sample on $x$.

Interestingly, adversarial learning competitions can also be organized as interactive benchmarks, where particiants' models can attack and defend against each other. Two designs are possible: the **sequential design** where the competition unfolds in distinct stages or phases, and the **simultaneous design** where challenges run both phases concurrently. Examples of sequential adversarial challenges include the ASVSpoof Challenge (Yamagishi et al., 2021; Liu et al., 2023) and the Data Anonymization and Re-identification Challenge (DARC) (Boutet et al., 2020). Examples of sequential adversarial challenges include the Hide-and-Seek Privacy Challenge (Jordon et al., 2020) and the Privacy Workshop Cup (Murakami et al., 2023).

One of the most famous adversarial learning competition is NeurIPS 2017 Adversarial Attacks and Defences Competition organized by Google Brain. This competition is consisted with 3 tracks: 1) non-targeted black-box attack; 2) targeted black-box attack; 3) defense against adversarial attacks. In each track, the participants submitted their models, then the submitted model was given a set of images (and target classes in case of targeted attack) as an input, and had to produce either an adversarial image (for attack submission) or classification label (for defense submission) for each input image Kurakin et al. (2018). The performance of attack models were measured by the average accuracy of victim models, and the performance of defense models were measured by their average accuracy against attack models. 91 teams participated the track 1), 65 teams participated the track 2), and 107 teams participated the track 3).

In IJCAI 2019, Alibaba Group organized an adversarial learning competition including 3 tracks: targeted attack track, non-targeted attack track and the defense track ijc (2019). In this competition, 110,000 pictures of goods from 110 commodity categories are published as training and test sets. In the attack tracks, the submissions were required to attack 5 defense models, then the average disturb norms on these 5 models were used to evaluate the performance of submissions. In the defense track, the submissions were also tested by 5 different attack models, then the average disturb norms of adversarial samples were disputed as the score of submissions. 2519 teams participated this competition to compete a total prize of 39,000 dollars. The teams from USTC won the championship of

defense track, the teams from Southeast University won the championship of target attack track, the teams from Guangzhou University won the championship of non-target attack track.

In KDD 2020, biendata and zhipu.AI organized a competition about the adversarial learning on graph data bie (2020). In particular, this competition is focus on the evasion attack and defense on the citation network de Solla Price (1965). The citation network is a kind of academic graph where academic papers are the nodes and citations are the directed edge. This graph is an important tool which can help researchers to analyse the cite relation of each paper and evaluate the impact of papers. Preventing the attack against the citation network (for example, manipulating citations Chawla (2019)) This competition included 2 phases, in which 543,486 nodes were training set and 50,000 nodes were test set. In the first phase, organizers provided a graph with 593,486 nodes and 100 features on each node. The participants were required to submit a black-box attack model to mislead the organizer's classifier by adding no more than 500 nodes. The performance of attack model was evaluated by the decrease on accuracy of organizer's classifier. In the second phase, each team submitted an attack model and a defense model trained on a similar but different dataset with the one of first phase. Then, the organizer matched all attack models and all defense models. The score of defense model was disputed by the average accuracy on each match, and the score of attack model was disputed by the average error rate on each match. The final score of each team was the average score of its attack model and defense model. 608 partcipants from 511 teams joined this competition to compete a total prize of 20,000 dollars.

Here we provide some tips for participants who wants to get good performance in adversarial learning competitions. For the evasion attack competitions, it is a good idea to combine the attack strategies with the domain knowledge. It is also important to have a well-designed adversarial loss function. For the defense competitions, there are two aspects in which the defense model can be improved. In the feature aspect, feature processing technology such as feature denoising Xie et al. (2019) and feature transformation Song et al. (2020) can help to improve the adversarial robustness of submissions. Some novel models such as topology adaptive model Du et al. (2017) can also be used to construct the adversarial robust model.

## 6 Use of confidential data

Confidential data may include sensitive information such as personal data, financial data, or trade secrets, and it is important to ensure that this data is handled in a secure and ethical manner. There are several concerns associated with using confidential data in machine learning competitions, including the need to protect the data from unauthorized access, and the need to comply with relevant laws and regulations. Confidential data holds a great importance in many applications, both in scientific and industrial contexts. Some examples include finance, healthcare, and human resources. Using confidential data in crowd-sourced benchmarks is a challenge in itself, but can be highly beneficial by enabling innovation in critical fields.

In this section, we present two different protocols for handling confidential data: **replacing the data by synthetic data**, and **running the participants' models blindly on the real data**. These two protocols, with their advantages and drawbacks, make it possible to crowd-source research on private data without compromising confidentiality.

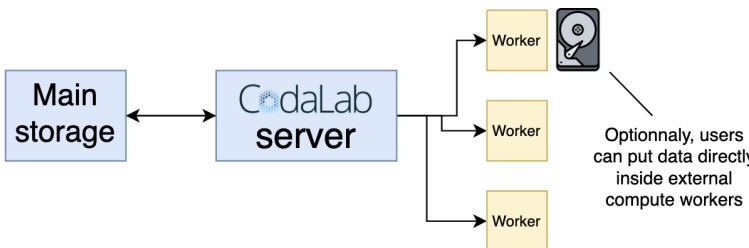

Figure 8: Confidential data can be put directly inside organizers' compute workers, externally from the main servers of the platform.

### 6.1 Synthetic data

In order to propose a task based on confidential data to the participants without exposing the private data, one approach is to train a generative model to replicate the dataset. Synthetic data can then be generated from the model, and used to simulate the task without disclosing the actual dataset. This approach raises two antagonistic issues: in one hand, the synthetic data must resemble the original data to ensure the problem remains relevant and connected to the real world; on the other hand, the generative model must not leak any real data points. We have developed metrics to evaluate generators *utility* and *privacy* (Yale et al., 2019, 2020) (presented in the chapter 4).

The limitation of using synthetic data is the potential trade-off between *privacy* and *utility*. The *utility* of artificial data can be evaluated by deploying it in real-world scenarios and verifying that model outcomes are consistent with those achieved using real data.

We applied this concept in "To be or not to be", referenced in Pavao et al. (2019), a challenge designed to instruct health students. The task is to predict the survival or decease of patients in intensive healthcare units, based on tabular medical records. The source of the data is the MIMIC-III dataset, which consists in both numerical and categorical variables describing thousands of patients, such as age and blood pressure. Given the inherent confidential and sensitive nature of this data, it is subject to access restrictions. We generated a synthetic dataset using a Wasserstein Generative Adversarial Network (WGAN) (Goodfellow et al., 2014; Arjovsky et al., 2017) model. The resultant challenge continues to be used in Rensselaer Polytechnique Institute to train health students[2].

### 6.2 Blind access to the data

The second approach for utilizing private data is to blindly execute participants' models on the real data. Two mechanisms are in play to benchmark the participants' solutions despite the private nature of the data: **code submission**, and **storing the data inside the compute workers**, as exposed in Figure 8. This way, only the uploaded models can read the data, it remains completely hidden from the participants. We implemented this feature to *CodaLab Competitions*. This is particularly interesting since, as shown in the chapter 11, organizers can link their own machines to the platform as external compute workers, ensuring a complete control over the data security. We employed this approach in the Paris Region AI Challenge 2020 (Pavao et al., 2021)

---

[2]https://codalab.lisn.upsaclay.fr/competitions/3073

| Protocol | Data | Multiple tasks | Code submission | Interactive design |
|---|---|---|---|---|
| Supervised learning | ✓ | | | |
| AutoML | ✓ | ✓ | ✓ | |
| Metalearning | ✓ | ✓ | ✓ | |
| Time series | ✓ | | | *depends* |
| Reinforcement Learning | *depends* | | ✓ | ✓ |
| Confidential data | ✓ | | *depends* | |
| Adversarial challenges | *depends* | | *depends* | ✓ |

Table 1: Characterization of the challenge protocols presented in the chapter, indicating the specific criteria that are mandatory (✓), and highlighting those that are possible depending on the design of the challenge (*depends*).

A sample of artificial data, as well as documentation and baseline methods, should be provided to help the participants building their methods despite the constraints associated with not being able to access the dataset directly. Having extensive output logs can also helps the participants to navigate through the problem despite of the blind testing. However, it is advised to limit the size of the output logs to avoid the leakage of the sensitive data. The security of external workers is ensured because the computer workers are owned by the organizers, and *CodaLab Competitions* platform cannot read them. The main limitation of this approach is that it is harder for the participants to work without direct access to the data, making it more difficult to reach the same performance level.

## 7 Conclusion

In this chapter we analysed the features and special designs of various type of machine learning competitions (adversarial learning, automated machine learning, etc.). We believe that the analysis in this chapter can help both organizers and participants, and also offers reference and inspiration about competitions of novel machine learning paradigms in the future.

In this chapter, we focused on examining the design specificity inherent in competitions and benchmarks in machine learning. We illustrated various experimental designs: supervised learning, AutoML and metalearning, time series analysis, reinforcement learning, the use of confidential data and adversarial challenges. The main characteristics and differences between these designs are outlined in Table 1. A common thread of most of these protocols is the necessity for participants to submit their model's code to the platform for evaluation. This resonates with the recommendation to use code submissions, both allowing complex evaluation procedures and improving the reproducibility and the validity of the evaluation. Interactive designs are at play in reinforcement learning, where algorithms interact with a dynamic environment; in adversarial challenges, where competing algorithms engage with one another; and occasionally in time series prediction tasks, where datasets are regularly augmented with new observations, allowing previously used testing data to become part of the training set for future iterations.

It is also interesting to note that artificial data holds potential utility in certain challenge designs. For tasks where the ground truth is almost exclusively artificial data, or when emulating real data that is confidential, synthetic datasets are beneficial. The latter can also be addressed with real data, by employing blind-testing methods, ensuring participants cannot access confidential datasets. More generally, synthesizing artificial datasets can be beneficial for tasks lacking a ground truth, such as in unsupervised learning, since the synthesis rules can be precisely known by the organizers. Indeed, synthetic data in machine learning competitions can offer a controlled environment to evaluate algorithms, with the advantage of generating diverse and challenging scenarios that resem-

ble complex real-world data distributions. Moreover, using artificial data, organizers can control the task difficulty, generate large datasets, address data imbalance, and reduce data collection cost. Additionally, in scenarios like reinforcement learning, where agents must learn from interaction within an environment, synthetic data provides an endless landscape of tasks for testing the robustness and adaptability of algorithms, as seen in competitions such as the *AI Driving Olympics* (Zilly et al., 2019). The main drawback of this approach is the potential *reality gap* between artificial and real data.

Adversarial learning, focusing on attack and defense algorithms, allows to explore the boundaries of the strengths and weaknesses of existing models. Adversarial learning competition can either focus on attack, on defense, or on both using an interactive design.

Given the diverse and rapidly evolving nature of the field of machine learning, a comprehensive enumeration of all possible design features and evaluation criteria is impossible. For instance, competitions centered on one-shot learning might evaluate the ability of models to generalize from minimal data, while those focusing on fairness could prioritize unbiased predictions across diverse demographic groups. In the domain of real-time processing, the emphasis might shift to algorithmic speed and responsiveness. Tasks involving multi-modal learning demand the integration of information from varied data sources like text, images, and audio. Meanwhile, resource-constrained competitions challenge participants to optimize the model performance with tight computational or memory budgets. However, the methodologies and approaches outlined here can serve as references for future competitions, particularly those in emerging paradigms such as automated machine learning or adversarial challenges.

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

# Practical issues: Incentives, community engagement and costs

**Magali Richard**  MAGALI.RICHARD@UNIV-GRENOBLE-ALPES.FR
*TIMC*
*UMR 5525, Univ. Grenoble Alpes, CNRS*
*F-38700, Grenoble, France*

**Yuna Blum**  YUNA.BLUM@UNIV-RENNES1.FR
*IGDR*
*UMR 6290, ERL U1305, Equipe Labellisée Ligue Nationale contre le Cancer, Univ Rennes, CNRS, INSERM*
*Rennes, France*

**Justin Guinney**  JGUINNEY@UW.EDU
*Tempus AI, Inc.*
*Chicago, IL 60654, USA*

**Gustavo Stolovitsky**  GUSTAVO.STOLO@GMAIL.COM
*DREAM Challenges*
*New York, NY, USA*

**Adrien Pavão**  ADRIEN.PAVAO@GMAIL.COM
*Université Paris-Saclay, France*

**Reviewed on OpenReview:** *https://openreview.net/forum?id=aATSw1zwOV*

## Abstract

Each organization of competitions and benchmarks involves a large number of practical problems, such as obtaining sufficient financial support or recruiting participants through appropriate incentives and community engagement. In addition to defining scientific tasks, preparing data and creating challenges, a very important practical administrative organization remains to be achieved. Indeed, cost assessment, corresponding requests for financial support and adequate publicity are key factors for successful organization of the competition. In addition, a good understanding of the incentives that lead participants to engage in a given challenge is fundamental for effective practical organization success. In this chapter, we will cover these topics and give some practical tips and examples for overcoming the "challenge" of organizing the challenges.

**Keywords:** practical issue, cost, publicity, management

This chapter provides a comprehensive guide to organizing successful scientific competitions, addressing both strategic and practical aspects of challenge organization. We begin by exploring participant motivations and incentives, offering insights into what drives researchers, students, and professionals to engage in scientific challenges. The chapter then delves into community building and outreach strategies, detailing effective methods for recruiting participants and disseminating challenge results within the scientific community. The final sections address the practical aspects of challenge management, including detailed breakdowns of financial and human resource requirements, along with guidance on securing funding sources. Throughout the chapter, we provide concrete examples and actionable recommendations drawn from successful competitions across various scientific domains.

The recommendations presented in this chapter stem from a multi-faceted approach to understanding competition organization. While our primary insights derive from extensive practical ex-

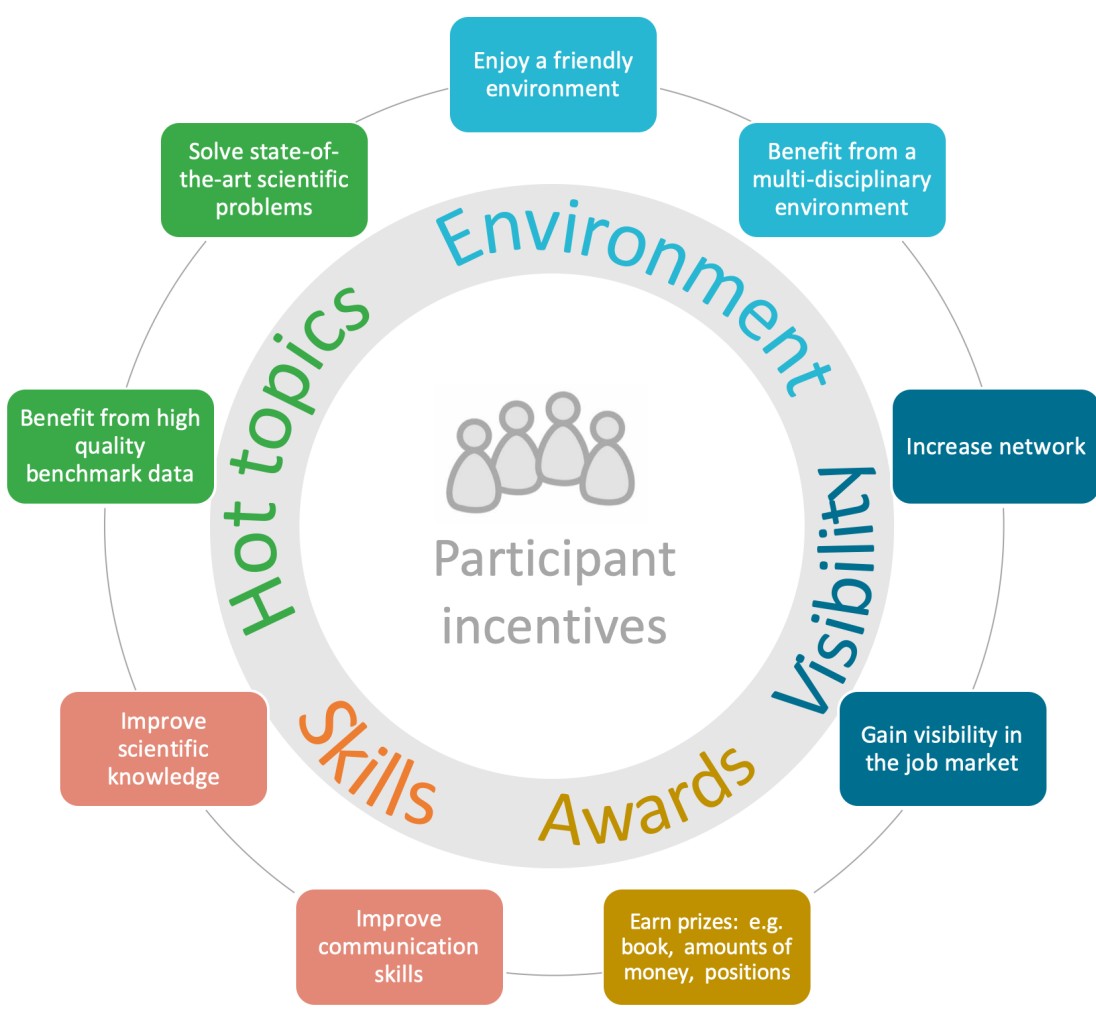

Figure 1: The incentives for participating in a challenge.

perience in organizing scientific competitions, we have strengthened these empirical observations through systematic analysis of documented outcomes from past competitions across various scientific domains. This analysis is complemented by structured feedback collected from both previous participants and experienced organizers, providing valuable perspectives on what contributes to competition success. Furthermore, we have aligned our practical recommendations with current research in the field, particularly regarding best practices in data handling, participant engagement, and competition design. This combination of practical experience, documented evidence, and academic research provides a robust foundation for the guidelines presented throughout this chapter.

# 1 Incentivizing participation

How to incentivize participants to work on complex problems is a key feature of challenge organization. In this section, we review various types of motivations (Figure 1), from a participant perspective.

## 1.1 Skills: Knowledge acquisition, communication, education

Traditional university programs in Artificial Intelligence are evolving rapidly, trying to meet the new needs of students, especially on their ability to work collaboratively while improving their scientific knowledge on data mining. Data challenges are mainly based on a coopetitive model, which has the advantage of responding to this dual motivation. Coopetition ((Brandenburger and Nalebuff, 2011) is an active learning pedagogical approach based on the combination of a strategy of competition, where students compete for the best result, and cooperation, where students collaborate for a mutual benefit. Coopetition-based data challenges have the advantage of simultaneously offering two types of learning. On the one hand, this gives a participant a solid methodological training on the scientific question addressed, thanks to the sharing of knowledge between professors and students, but also between the students themselves. On the other hand, these approaches allow students to acquire new skills in collaboration, communication and networking. For more details, please refer to chapter 9: Competitions and challenges in education.

Educational data challenges can be organized into teamwork, recruiting participants from different backgrounds (academic and cultural), with a scientific preparation that can range from minimal information about the challenge before starting to full preparation through a series of dedicated conferences. To meet the expectations of the students, a key factor is the will of the organizer to build a "friendly environment" which will help to boost the motivation of the students and their self-esteem, and to focus more on the process itself than on the results and objectives. Building multidisciplinary teams with different scientific expertise and focusing on real problems are important aspects in the organization of educational challenges. It is also important to provide an environment where participants can communicate with their team members, other teams, and teachers. Ultimately, setting the right reward and price is a major motivator for winning student buy-in (Abernathy and Vineyard, 2001).

Finally, organization of competitions itself can be used as a pedagogical tool. Designing such task is complex and can be, in some regards, more interesting than solving it (Pavao et al., 2019).

## 1.2 Hot topics: Scientific crowdsourced benchmarking

The quintessential challenge revolves around an existing quantitative standard or benchmark, and seeks to improve upon state-of-the-art. One of the more longstanding benchmark initiatives is the Critical Assessment for Structural Proteins (CASP) that asks participants to predict protein structure (folding) from protein sequence. Groups who specialize in this domain are naturally incentivized to compare their approach in the structured and objective format of a data challenge in the hope that their method out-competes other approaches and can therefore become a new standard in the field (Bender, 2016). CASP is now recognized within the protein structure community as the *de facto* forum for assessing algorithms, and is therefore as much an incentive as a mandate for formal recognition with the community. This incentive generalizes to all specialties, including image recognition (e.g. MNIST (Madry et al., 2019), ImageNet (Russakovsky et al., 2015)), gene identifi-

cation and function prediction (e.g. RGASP (Steijger et al., 2013), CAFA (Radivojac et al., 2013)) or translational research in biomedicine (Saez-Rodriguez et al., 2016).

Any published AI algorithm is expected to include a formal performance comparison against state-of-the-art methods. No good data-driven approach could emerge without good quality, well curated data. This task can be cumbersome and require a great deal of work to assemble and prepare benchmark datasets. Depending on the type of data, data acquisition and/or generation can be very time-consuming and costly (see cost section below). Consequently, a natural perk of a scientific data challenge is that the work involved to generate and prepare a benchmarking dataset is managed by the challenge organizers. Therefore, AI competitions offer a playground with data that are usually costly and complicated to generate. Access to high-quality datasets in machine learning remains an ongoing challenge (10.1007/s00778-022-00775-9). We believe that providing access to such datasets serves as a strong motivation for participants seeking to develop cutting-edge methodological approaches to address complex scientific problems..

Recurrent challenges also present the advantage or keeping people on a regular schedule, as they expect the challenge to come and reserve time for it. As foor a classic scientific event, it provides participants the opportunity to expand their professional network and to start new collaborations with people working in the same research field or people from different disciplines gravitating around the same topic. Finally, data challenges remain the best functioning way of implementing coopetitions: people compete and get credit for winning, then they share their solution publicly and the community can move together to the next step.

### 1.3 Environment and awards

One appealing aspect of the challenges is the spirit of games. This translates into a friendly yet competitive environment along with rewards. It is not unusual to gather common participants on different challenges. A passion to participate in this type of competitions can develop, along with the excitement of witnessing the evolution of the social community, particularly on commercial platforms like Kaggle. The rewards can be of various nature going from small prizes (e.g. book) to high amounts of money (e.g. 1 million dollars, Salesforce 1 Hackathon) or even positions in companies. Large awards may naturally attract more participants, but this must be balanced with the context of the challenge and the scientific problem being addressed. In other words, factors such as feedback, non-monetary recognition, and opportunities for knowledge advancement should also be considered.

### 1.4 Visibility, career and recruitment

Challenges are opportunities for participants to showcase their various skills to recruiters and even get a position at the end. A growing number of organizations are adopting modern hiring practices such as challenges to find best candidates. Recruiters use this tool to assess candidates' technical and behavior skills. Challenges have indeed the great advantage of evaluating many different criteria at the same time. Companies can assess technical competencies such as problem solving skills, time management and innovations. They can also assess the behavioral skills they value, such as communication, openness to diversity and leadership.

The implementation of a challenge allows recruiters to define certain expectations towards the evaluated candidates (candidates gain insight into the work culture of their future employer), while verifying if their personality corresponds to the company's fundamental values. One of the diffi-

culties in recruitment is that many companies still follow long selection processes that waste time and interest for both candidates and recruiters. To overcome this problem, challenges can be used to evaluate candidates in a short period of time and a friendly environment, where they can demonstrate real-time expertise. It can also serve as a pre-selection process that will also save time for recruiters.

Interestingly, challenges can bring together a larger number of candidates from more diverse backgrounds than traditional recruiting. Organizers can build a portfolio of interesting candidates for present and future positions, without necessarily limiting themselves to the winners of the challenge. For instance, Kaggle, one of the leading challenges platform acting as a recruiting tool, usis a performance tracking system to evaluate participants[1]. Some companies even sell expertise from Kaggle Grandmasters[2]. Besides, challenges are also an excellent way to increase brand awareness. They can be used as a marketing tactic for big companies to reinforce their leadership in their field. Smaller companies can also increase their visibility though challenges and attract more applicants for a position.

Finally, in addition to recruiting new talent, challenges allow companies to bring innovative solutions and ideas to technical problems. Based on the clear success of challenges in the recruitment process, we can easily expect their increase in the upcoming years.

---

1. `https://www.kaggle.com/progression`
2. `https://h2o.ai/company/team/kaggle-grandmasters/`

> **Practical tips and resources to optimize incentivization**
>
> - Define your working plan and your objectives[a]
>
> - Carefully prepare benchmarking datasets (see Chapter 3 on data preparation).
>
> - Set up a website to collect a list of interested people[b].
>
> - Bring together an expert steering committee
>
> - Provide good educational material together with the challenge (i.e. a good starting kit, white paper).
>
> - Make yourself available during the challenge to answer questions.
>
> - Be responsive to questions on the forum.
>
> - For a recurrent challenge, provide open-source previous winning solutions.
>
> - Organize good publication venues (see details and examples in section 12.2 Community engagement)
>
> - Associate with established conferences (see details and examples in section 12.2 Community engagement)
>
> - For education challenges, you can find inspiration on existing education challenges on open-source platforms such as RAMP[c], or Codalab[d]
>
> ---
>
> a. 10 tips here: http://www.chalearn.org/tips.html
> b. see e.g. https://l2rpn.chalearn.org/
> c. https://ramp.studio
> d. https://codalab.lisn.upsaclay.fr/

## 2 Community engagement

Mechanisms for engaging and disseminating a competition towards a targeted community are complex and highly dependant on the scientific field. In this section, we try to review general aspects of community engagement that could help challenge organizers to properly define their strategy. See Figure 2 and Table 1 for a review of community engagement strategies and examples of recent competitions.

### 2.1 Organization of the challenge

The community that will engage in a specific competition will depend on several key aspects of defining the challenge. First, the organizers should define an optimal number of participants and implement the maximum number of participant (if any). Large open competitions have the advantage of ensuring visibility and optimizing scientific production (in the case of crowdsourced benchmarking for example) while smaller competitions will promote communication between participants (more adapted to challenges aiming at educational results). Then they have to determine

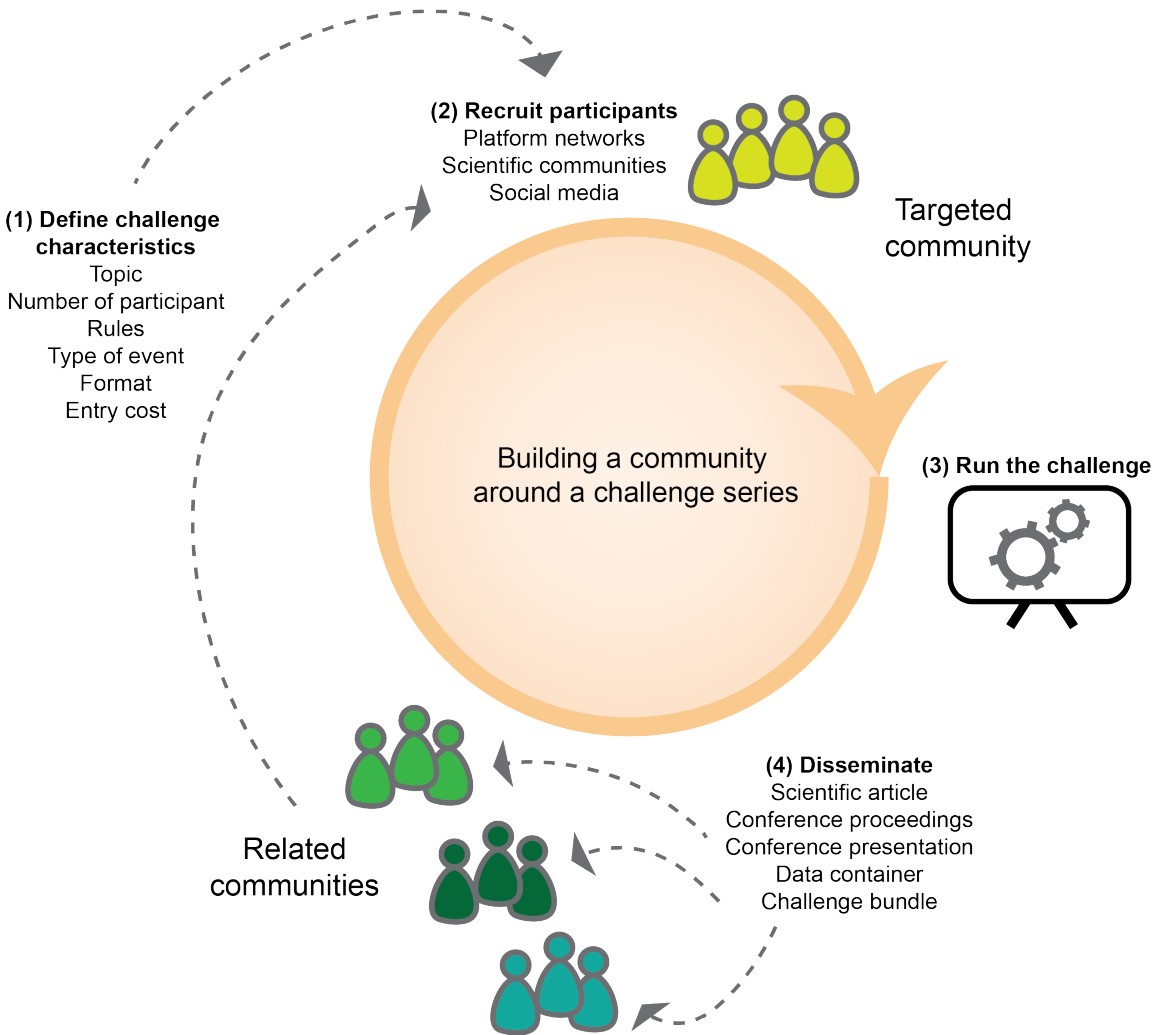

Figure 2: The process of engaging a community

an entry cost: is it easy to participate in the competition? The entry cost depends on several factors: clarity of the rules, specificity of the tasks, size of the dataset, computational resources required to run the methods... All of this will have an impact on the participants who will enter the competition and indirectly define the target audience. Finally, the organizers should established what the format of the competition will be: online events will increase the chances of getting a large pool of participants while in-person events (e.g. at dedicated schools or at scientific conferences) are more suitable for collaborative team work. Once all these parameters are specified, organizers can adapt their communication strategy accordingly and start communicating through dedicated channels, such as the scientific communities mailing list, the digital challenge platform networks and the social media.

## 2.2 Ensuring diversity and inclusion

A crucial aspect of organizing scientific competitions is ensuring Equity, Diversity, and Inclusion (EDI). Challenge organizers must proactively work to make their competitions accessible and attractive to participants from diverse backgrounds. This includes considering participants who may be traditionally underrepresented due to their gender, race, socioeconomic status, or neurodivergence. Practical measures include offering flexible participation options, such as remote participation possibilities and adjustable deadlines to accommodate different time zones and work constraints. Financial accessibility should be addressed through measures like reduced registration fees for students and participants from low-income countries, or travel grants for in-person events. Ideally, the competition platform and documentation should be designed with accessibility in mind, ensuring compatibility with screen readers and providing materials in multiple formats. Additionally, organizers should establish clear codes of conduct and communication guidelines that promote respectful interactions and create an inclusive environment. The selection of challenge topics, datasets, and evaluation metrics should also be examined for potential biases that might disadvantage certain groups. Building a diverse organizing committee can help identify and address potential barriers to participation early in the planning process. Regular feedback from participants about accessibility and inclusion can help refine these measures over time.

## 2.3 Challenge output dissemination

The dissemination of the data challenge can take several formats (complementary and not exhaustive) and should match the following question: how would it serve the targeted community?

Participatory benchmarking competitions generally result in scientific publications (see examples (Creason et al., 2021; HADACA consortium et al., 2020; Marbach et al., 2012; Eicher et al., 2019; Marot et al., 2021; Le et al., 2019)) which will be of use to the community. Offering authorship to competing teams, along with participation in manuscript design and writing, is also a strong incentive that will provide international visibility and recognition to participants. Organizers might try to connect with high-profile journal editors ahead of the challenge organization to discuss the possibility of publishing the competition outcome. Depending on the scientific field of the competition, publications can take various form, such as scientific articles, contributions to special issues, conference proceedings, or even books. Best performing teams can also be offered the ability to present their solution in international scientific conferences (e.g., since 2008 all best performing teams in the DREAM Challenges present at the yearly "RECOMB/ISCB RSGDREAM" conference). In addition to an article describing the results of the competition, a challenge built on the data to modeler model (Guinney and Saez-Rodriguez, 2018) could also result in publishing the benchmark dataset along with a container providing a reproducible and continuous benchmark (e.g. a dedicated docker container). Competition data can then be re-used by research scientists as gold standard for new computational methods that will be developed in the future. Challenge organizers may also consider giving open access to their challenge design and templates, especially regarding educational challenges, so that these competition can be massively disseminated to various universities at no cost.

Challenge output and dissemination strategy differ a lot according to the competition organizers and environments. Academic competitions massively rely on the open science framework, encouraging participants to submit their code under an open source license (ex: L2RPN, DREAM challenges). On the opposite, private companies are often motivated by solving an theoretical and

methodological obstacle in order to further develop private commercial solutions that will be put on the market. Such organizers may be more inclined to follow a 'private output' model where participant surrender intellectual property of their findings in exchange for earning money prizes.

| COMMUNITY ENGAGEMENT | | | | | |
|---|---|---|---|---|---|
| Name | Field | Year | Platform | Number of participants | Dissemination |
| TrackML Particle Tracking Challenge | Physics | 2018 | Kaggle | 739 participants | IEEE WCCI competition (Rio de Janeiro, Jul 2018) and NIPS competition (Montreal, Dec 2018) |
| LAP series | Computer Vision | 2013-22 | CodaLab | more than 300 teams | Springer Series on Challenges in Machine Learning, ECCV, IEEE TPAMI, JMLR, IJCV, PAA, CVPR |
| Tumor Deconvolution | Health | 2019-20 | DREAM | 38 teams | 2019 RECOMB/ISCB Regulatory and Systems Genomics, Nature Communications |
| AutoDL series (6 competitions so far) | Automated ML | 2019-21 | CodaLab | more than 300 teams | ECML/PKDD, ACML, NeurIPS, IJCNN, WAIC, IEEE TPAMI |
| Digital Mammography | Health | 2017 | DREAM | 126 teams | RECOMB/ISCB Regulatory and Systems Genomics, JAMA Netw Open. |
| L2RPN | Energy | 2020 | CodaLab | more than 300 participants | NeurIPS, ArXiv |
| Challenge AI for industry | Aeronotic | 2021 | CodaLab | 10 teams | |
| HADACA series (3 competitions so far) | Life sciences | 2018-24 | CodaLab | 150 participants | BMC bioinformatics, JOBIM |

Table 1: Table of communities engagement

As a complement, a non-exhaustive list of conferences that have call for competitions, or can offer workshops and/or proceedings, as well as journals that can welcome competition result publication :

- *Conferences and workshops*: ESANN, ICMLA, WCCI (IJCNN, CEC), ECML/PKDD (Discovery challenges), KDD (KDD cup), CVPR, ECCV, ECML/PKDD, ICPR, ICDAR, IEEE international conference on big data, IEEE International Conference on Automatic Face and Gesture Recognition (FG), ACM SIGIR Forum, NeurIPS dataset and benchmark track, NeurIPS competition track, Workshops @ NeurIPS, ICML, AAAI, CVPR, ICCV, Workshop on Semantic Evaluation, etc.
- *Book series*: CiML Springer series, etc.
- *Journals and pre-prints*: International Journal of Forecasting, International Journal of Information Retrieval Research (IJIRR), IEEE Journal of Biomedical and Health Informatics, IEEE Access, Machine Vision and Applications, IEEE TPAMI, Nature methods, Nature com-

## 3 Costs, human labor and resources

Depending on the model chosen by the organizers, various costs will be associated with a competition organization. To mitigate the problem of financing a competition, diverse sponsors, private companies or academic institutions can be involved. See Figure 3 and Table 3 for a review of costs and resources associated with recent competitions. Complementary to this section, "Chapter 2: Challenge Design Roadmap" offers guidelines and case studies for developing a robust plan for challenge design.

---

**An example of challenges costs: the L2RPN challenge / NeurIPS 2020**

- **Research field**: Energy and environment.

- **Challenge platform**: Codalab[a].

- **Duration of the challenge**: 4 months.

- **Number of participants**: 300.

- **Data generation, access and curation: costs and resources description** : 70,000 euros.

- **Challenge engineering: costs and resources description**: 120,000 euros.

- **Challenge design, scientific expertise: costs and resources description**: 170,000 euros.

- **Prices, travel, conference organization (approximate evaluation of costs)**: 30,000 euros.

- **Challenge governance (cost evaluation of legal, ethics and data privacy costs)**: none.

- **Dissemination**: RTE, Google Research, University College of London, EPRI, IQT Labs. Chalearn.

- **Sponsors**: PMLR[b] & ChaLearn[c]

---

[a]. https://competitions.codalab.org/competitions/25426
[b]. https://arxiv.org/abs/2103.03104
[c]. https://l2rpn.chalearn.org/

---

### 3.1 On overview of the requirements and associated costs

PLATFORM AND REGISTRATION SYSTEM

Several digital platforms can support challenge organization (see chapter 5 for different models of challenge platforms). Defining the platform should be a starting point of challenge organization, as open-source projects such as CodaLab or commercial challenge platforms such as Kaggle will provide very different resources (technical support, engineering manpower dedicated to the compe-

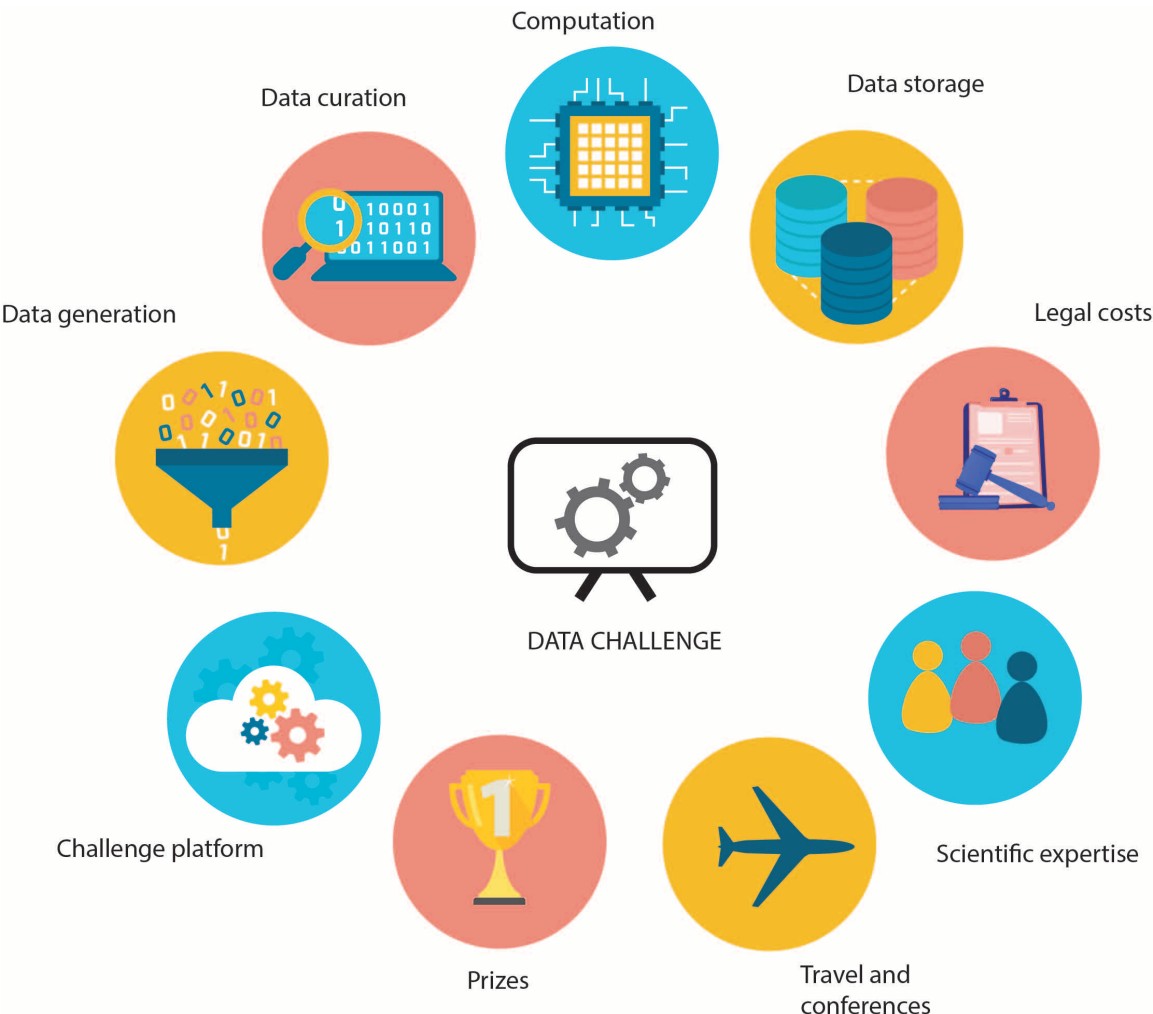

Figure 3: Costs of data challenge organization. Pictures adapted from open work on freepik: macrovector, alvaro_cabrera, visnezh & vectorjuice.

tition...) and associated costs. Please refer to Chapter 5 for more details on the different services provided by each platform.

### DATA GENERATION, ACCESS AND CURATION

High-quality, well-curated data is fundamental to competition success. Recent research, particularly the work of Mougan et al. (2023), has provided comprehensive frameworks for data handling in scientific competitions. A high-quality dataset requires careful planning across multiple dimensions: from initial requirement analysis that clearly establishes the dataset's purpose, through implementation considerations such as sample size and data balance, to thorough documentation and annotation. Additionally, a robust data management plan is essential to ensure data integrity and accessibility throughout the competition (for detailed guidelines on dataset development, see Chapter 4). This

structured approach to data preparation helps ensure that the competition's scientific objectives can be effectively addressed while providing participants with reliable resources for developing their solutions. General cost evaluation of data generation is complicated because it is highly variable depending on the scientific discipline involved. Data generation has always a cost, but this cost can bee supported by different players of the competition (sponsors, private companies, organization committee, care providers, health insurance, etc). This costs also depends on the data type, size and accessibility. Good quality data also relies on the willing of organizers to work in synchronisation with the global efforts for technical standardization and ethic responsible data sharing, e.g. Global Alliance for Genomics and Health or FAIR principles for data management and stewardship (Wilkinson et al., 2016; Cabili et al., 2018).

## GOVERNANCE AND LEGAL COSTS

Competition governance strategy should also include legal counseling costs, that will ensure that the data storage and sharing concept complies with national and international legal requirements. In particular, usage of identifiable personal data (such as patient clinical data) is a complex and significant legal and data protection challenge (Nicol et al., 2019). Moreover, rules for awarding prizes and travel grants should be clearly defined, this includes definitions of:

- jury's composition (committee of experts)

- criteria of evaluation (e.g. relevance, usefulness, novelty, etc.)

- challenge submission process

- intellectual properties

- exclusion and appeal procedures

- control of the use of funds and goods, including prices

- privacy policy

- errors, frauds and breaches of rules mitigation plan

## COMPUTATION AND STORAGE

The digital data challenge platforms rely on cloud computing services to run and evaluate models. Access to these services can be externalized (such as Google Could Platform, Openstack, IBM Cloud or Amazon web services) or provided internally using the computing infrastructure of the challenge organizers. Depending on the competitions, the problem to solve and the type of data, the required resources vary a lot. For instance, in the case of code submission, it is important to estimate well the number of participants, and sometimes to limit the entries by setting a hard threshold. Indeed, code submission offers many advantages (controlled environments, confidential data, good sharing of the resources among participants, etc.) but is computationally very demanding. Thus, the organizers must accurately estimate the computation time of the expected methods as well as the type of computing units to use ((Ellrott et al., 2019)), knowing that donation of cloud units from Google, Azure and Amazon are relatively easy to obtain. Some platform such as Codalab can be coupled with such cloud services, via the use of compute workers. Finally, they need to decide accordingly whether they wish to offer computational services (allowing code submission) or ask participants to provide their own computational resources (only allowing the submission of results).

SCIENTIFIC EXPERTISE, CHALLENGE DESIGN AND ENGINEERING

Bringing together an expert steering committee is a key factor to ensure that the issue raised by the competition corresponds to the needs of the community, and that the data will be used correctly to ask the right question. These two points are essential to ensure community engagement and the quality of the competition. Code development is also an important factor to consider. In certain specific situations, building a dedicated application or a realistic environment to simulate the various tasks of a competition can demand significant effort, including extensive research and substantial engineering manpower prior to the competition. For instance, L2RPN competition series required the generation of a dedicated framework and the generation of synthetic data with several people working on the project for over a year (cost of $\sim$200k€). Once the competition is completed, manpower is also needed to analyse the results, summarize, and disseminate the challenge outcomes.

PRIZES, TRAVEL AND CONFERENCE ORGANIZATION

Reward costs should be included in the challenge budget. Prizes can be an important incentive to recruit participant (see section 1). In case of in-person events, travel and conference organization costs should be considered. This can include speakers invitations, participation to the venue costs and travel grants for students. Competitions can be short (one week) or long (over several months), held remotely or in person, and may or may not be associated with an international conference (see "Part II : The best of challenges and benchmarks" for more examples of academic and industry competitions). All these elements must be taken into account when preparing the budget. For example, the HADACA challenges (Health Data Challenges) take place in the form of a one-week winter school in the French Alps, with around fifty participants. The total cost of organizing the event (including accommodation and meals) was €30,000 for the 2024 edition (HADACA3[3]). Example of costs to organized a one day workshop can be found in Table 2).

|   | Expense type | Estimated cost (EUR) |
|---|---|---|
| 1 | Invited speakers registration (4x$250) | 1,000 |
| 2 | Organizer travel expenses (3x$2000) | 6,000 |
| 3 | Lunch (catering) for 40*$50 | 2,000 |
| 4 | Dinner for invited speakers, winners, organizers (20*$50) | 1,000 |

Table 2: Conference or workshop organization for a total budget of 10,000 euros.

## 3.2 Person power

Person power is crucial in competition organization and should not be underestimated. While 3 provides an average estimation of person power required to organize a challenge, accurately estimating human resource needs remains one of the most challenging aspects of competition planning. These requirements often evolve throughout the competition lifecycle, with varying demands across different phases - from initial planning to final evaluation. Resource needs can fluctuate based on

---

3. HADACA3 website : https://hadaca3.sciencesconf.org/

unexpected technical challenges, participant engagement levels, or administrative complexities. A proven strategy to address this uncertainty is to establish a robust technical committee from the outset, comprising members with diverse expertise. This committee should include not only scientific experts but also professionals skilled in administrative tasks, accounting, publicity and communication, software development, data analysis, and reporting. Such diversity in expertise helps ensure that the competition can adapt to evolving demands while maintaining high standards across all aspects of organization. This distributed approach to human resources also provides redundancy and flexibility, allowing the organizing team to better handle peak workloads and unexpected challenges.

### 3.3 Resources: sponsors and grant agencies

As the global cost of competition organization grows along with the complexity of the data and tasks, proposal and grant writing to find money is essential. By leveraging institutional support and sponsors, organizers will achieve good quality challenges and ensure community participation. More and more universities and national funding agencies[4] or scientific societies[5] support competition organization. Building partnership with private companies[6] and involving collaborators in scientific consortium is also likely to be very helpful to reduce the financial barriers in organizing challenges.

## 4 Conclusion

Organizing a competition necessitates the dedication of a scientific committee, substantial time, and financial resources. It is imperative not to underestimate the level of commitment required for the successful execution of such events. However, potential organizers should not be discouraged. On the contrary, organizing a competition is a highly rewarding experience, and we encourage any aspiring organizer to undertake it. It's worth noting that competitions represent just one approach to collaborative science. Recent initiatives demonstrate the diversity of possible formats: from large-scale collaborative projects like BLOOM by BigScience [7], which brought together hundreds of researchers to create an open multilingual language model, to the development of innovative evaluation frameworks for language models. Furthermore, while this chapter has primarily focused on traditional competition formats, emerging approaches such as dynamic benchmarking offer promising alternatives to static competitions. These dynamic formats enable iterative data collection and model development, though they require specific design considerations to be implemented effectively."

This chapter is designed as a practical guide, and given the large number of competitions already held, newcomers to the field will find abundant examples to draw inspiration and ideas from. The recommendations and guidelines presented in this chapter are intended to serve as a theoretical framework, not as rigid constraints. The innovative nature of this field extends to the format and design of the competitions themselves, fostering a continuous environment of creativity and development.

---

4. For instance the University College of London, the National Research Agency in France, the ETH in Switzerland, or the EIT Health in Europe

5. National Science Foundation in the United States, the IEEE Computational Intelligenece Society, or the International Neural Network Society

6. Non-exhaustive list of potential sponsors: Google, Microsoft, Orange, Kaggle, Health discovery corporation

7. BLOOM: https://huggingface.co/bigscience/bloom

H]

| Task | Description | Hours |
|------|-------------|-------|
| 1 | **Finding/reviewing data.** | 50 |
| 2 | **Formatting data.** Preprocess and format the data to simplify the task of participants, obfuscate the origin, anonymize. | 100 |
| 3 | **Assessment.** Define a task and evaluation metrics. Define and implement methods of scoring the results and comparing them. | 50 |
| 4 | **Baseline software; starting kit.** Implement a simple example performing the tasks of the challenge. Prepare useful software libraries, make examples. | 100 |
| 5 | **Result formats and software interfaces.** Define the formats in which the results should be returned by the systems and how experimentation will be conducted during the challenge. | 50 |
| 6 | **Benchmark protocol.** Define the rules of the competition and determine the sequence of events. | 50 |
| 7 | **Web portal.** Implement on challenge platform the benchmark protocol allowing on-line submissions and displaying results on a leaderboard. | 25 |
| 8 | **Guidelines to participants.** Write the competition rules, document the formats and the scoring methods, write FAQs. | 50 |
| 9 | **Beta testing.** Organize and conduct tests of the challenge. | 25 |
| 10 | **Run the challenge.** Answer participants, attend to the platform (2h/week). | 100 |
| 11 | **Prepare the workshops.** Write proposals. Look for tutorial speakers. Select speakers. Create a schedule. Advertise. | 50 |
| 12 | **Competition result analysis.** Compile the results. Produce graphs. Derive conclusions. | 50 |
| 13 | **Reports.** Write reports on the benchmark design, the datasets, and the results of the competition. | 100 |
| 14 | **On-line result dissemination.** Make available on-line the competition result analyses, fact sheets of the competitors's methods, and the workshop slides. | 50 |
| 15 | **Prepare workshop proceedings.** Solicit papers, organize the review process, and edit the papers. | 100 |
| 16 | **Distribute prizes and awards.** | 10 |

Table 3: Evaluation of person power to organize a challenge (varies from challenge to challenge, should be estimated by the organizing team)

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
