# OpenReview forum: "AI Competitions and Benchmarks book - Practical issues and open problems"
_DMLR — Accepted by DMLR_

### Review · Reviewer_JmCi · 2024-12-18

**Recommendation:** 4
**Confidence:** 3

**Summary Of Contributions:**

The revised submission has successfully addressed the concerns raised in the previous round. The authors have incorporated necessary clarifications and improvements, including discussions on reproducibility, ethics, and practical issues. The grouping of chapters—covering competition platforms, hands-on tutorials, special designs, and practical considerations—provides a comprehensive and well-structured approach. I believe this version effectively resolves the prior gaps and meets the expectations set in the earlier feedback.

**Strengths:**

See Strengths And Weaknesses.

**Audience:**

Yes

**Broader Impact Concerns:**

Nan.

**Claims And Evidence:**

No.

**Datasets And Benchmarks:**

Yes.

**Extended Submissions:**

Yes.

**Limitations:**

No.

**Requested Changes:**

No.

**Strengths And Weaknesses:**

The submission is clearly structured, with chapters logically grouped to guide the reader through various aspects of designing and managing competitions and benchmarks.

---

### Review · Reviewer_Xjgy · 2025-01-07

**Recommendation:** 3
**Confidence:** 2

**Summary Of Contributions:**

This submission group provides general overview on what are the general competition platforms, how to create one competition, and discussions on the design and execution,

**Strengths:**

See Strengths And Weaknesses section

**Audience:**

Yes

**Broader Impact Concerns:**

I don't see concerns on this.

**Claims And Evidence:**

Yes, it provides clear citations and evidences to claims it make.

**Datasets And Benchmarks:**

It's not a dataset paper but on how to construct a benchmark for competition.
It does provide various examples in a wide range of domains on how the benchmark is set and how the competition is organized.

**Extended Submissions:**

It's not an extended submission.

**Limitations:**

See Strengths And Weaknesses section and Requested Changes.

**Requested Changes:**

1. The “Special Designs and Competition Protocols” section has some structural issues that could be refined. For instance, it places meta-learning and AutoML at the same hierarchical level, even though it acknowledges that “meta-learning is a sub-problem of AutoML.” Similarly, time series analysis and supervised training are presented in the same level sections, despite time series analysis has much overlap with supervised training.
2. At page 12, in 2024, the ARCathon has been rebranded as the Arc Prize and is now hosted on Kaggle. It would be helpful to update this information.

**Strengths And Weaknesses:**

Strengths:
1. The writing is clear, and the figures make the book chapters generally easy to follow.
2. The book’s structure is good. It provides a comprehensive overview of how benchmarks are constructed and how to organize a competition, which will be highly beneficial to many researchers.

Weaknesses:
1. Some concerns from the previous round of reviews remain unresolved. For instance, Action Editor V9mX pointed out that the proposal lacks a discussion on the important practical question of how to handle intellectual property generated through benchmark activities, as well as a historical perspective on significant long-running benchmark initiatives, such as TREC. These issues are still unaddressed.
2. There are also minor writing issues, as outlined in the “Requested Changes” section.

---

### Review · Reviewer_dq2F · 2025-08-30

**Recommendation:** 4
**Confidence:** 3

**Summary Of Contributions:**

I originally reviewed Chapter 13 of this bundle. It appears that this bundle contains the original version of the submitted chapter, as none of the changes promised by the authors in response to reviewers' comments appear to have been implemented in the attached file.
I am not sure if the above is an error or intentional, and if it affects only the chapter I originally reviewed or all chapters.

Please see https://openreview.net/forum?id=Trv971GZDm for the changes proposed by the authors of Chapter 13 in response to the reviewers' comments. I believe that these changes should be implemented before the book manuscript gets published.

Update: The authors provided a link to the updated version, which for some reason was not available in OpenReview. The updated version implemented some important parts of the review recommendations but left others unaddressed.

**Strengths:**

See https://openreview.net/forum?id=Trv971GZDm

**Audience:**

Yes

**Broader Impact Concerns:**

See https://openreview.net/forum?id=Trv971GZDm

**Claims And Evidence:**

See https://openreview.net/forum?id=Trv971GZDm

**Datasets And Benchmarks:**

See https://openreview.net/forum?id=Trv971GZDm

**Extended Submissions:**

See https://openreview.net/forum?id=Trv971GZDm

**Limitations:**

See https://openreview.net/forum?id=Trv971GZDm

**Requested Changes:**

See https://openreview.net/forum?id=Trv971GZDm

**Strengths And Weaknesses:**

See https://openreview.net/forum?id=Trv971GZDm